# Tuberculosis-associated IFN-I induces Siglec-1 on tunneling nanotubes and favors HIV-1 spread in macrophages

Maeva Dupont[1,2], Shanti Souriant[1,2†], Luciana Balboa[2,3†], Thien-Phong Vu Manh[4], Karine Pingris[1], Stella Rousset[1§], Céline Cougoule[1,2], Yoann Rombouts[1], Renaud Poincloux[1], Myriam Ben Neji[1], Carolina Allers[5], Deepak Kaushal[5#], Marcelo J Kuroda[5¶], Susana Benet[6,7], Javier Martinez-Picado[6,8,9], Nuria Izquierdo-Useros[6,10], Maria del Carmen Sasiain[2,3], Isabelle Maridonneau-Parini[1,2‡], Olivier Neyrolles[1,2‡], Christel Vérollet[1,2‡*], Geanncarlo Lugo-Villarino[1,2‡*]

*For correspondence:
verollet@ipbs.fr (CV);
lugo@ipbs.fr (GL-V)

†These authors contributed equally to this work
‡These authors also contributed equally to this work

Present address: §Department of Infectious and Tropical Diseases, Toulouse University Hospital, Toulouse, France; #Southwest National Primate Research Center, Texas Biomedical Research Institute, San Antonio, United States; ¶Center for Comparative Medicine and California National Primate Research Center, University of California, Davis, Davis, United States

[1]Institut de Pharmacologie et Biologie Structurale, IPBS, Université de Toulouse, CNRS, UPS, Toulouse, France; [2]International associated laboratory (LIA) CNRS 'IM-TB/HIV', Toulouse, France; [3]Institute of Experimental Medicine-CONICET, National Academy of Medicine, Buenos Aires, Argentina; [4]Aix Marseille Univ, CNRS, INSERM, CIML, Marseille, France; [5]Tulane National Primate Research Center, Department of Microbiology and Immunology, School of Medicine, Tulane University, Covington, United States; [6]IrsiCaixa AIDS Research Institute, Department of Retrovirology, Badalona, Spain; [7]Universitat Autònoma de Barcelona, Barcelona, Spain; [8]University of Vic-Central University of Catalonia (UVic-UCC), Vic, Spain; [9]Catalan Institution for Research and Advanced Studies (ICREA), Barcelona, Spain; [10]Institut d'Investigació en Ciències de la Salut Germans Trias i Pujol, Badalona, Spain

Competing interests: The authors declare that no competing interests exist.

**Abstract** While tuberculosis (TB) is a risk factor in HIV-1-infected individuals, the mechanisms by which *Mycobacterium tuberculosis* (Mtb) worsens HIV-1 pathogenesis remain scarce. We showed that HIV-1 infection is exacerbated in macrophages exposed to TB-associated microenvironments due to tunneling nanotube (TNT) formation. To identify molecular factors associated with TNT function, we performed a transcriptomic analysis in these macrophages, and revealed the up-regulation of Siglec-1 receptor. Siglec-1 expression depends on Mtb-induced production of type I interferon (IFN-I). In co-infected non-human primates, Siglec-1 is highly expressed by alveolar macrophages, whose abundance correlates with pathology and activation of IFN-I/STAT1 pathway. Siglec-1 localizes mainly on microtubule-containing TNT that are long and carry HIV-1 cargo. Siglec-1 depletion decreases TNT length, diminishes HIV-1 capture and cell-to-cell transfer, and abrogates the exacerbation of HIV-1 infection induced by Mtb. Altogether, we uncover a deleterious role for Siglec-1 in TB-HIV-1 co-infection and open new avenues to understand TNT biology.

## Introduction

Co-infection with *Mycobacterium tuberculosis* (Mtb) and the human immunodeficiency virus (HIV-1), the agents of tuberculosis (TB) and acquired immunodeficiency syndrome (AIDS), respectively, is a major health issue. Indeed, TB is the most common illness in HIV-1-infected individuals, about 55% of TB notified patients are also infected with HIV-1, and about a fifth of the TB death toll occurs in HIV-1 co-infected individuals (WHO health report 2017). Clinical studies evidence a synergy between

these two pathogens, which is associated with a spectrum of aberration in immune function (*Esmail et al., 2018*). Yet, while progress has been made towards understanding how HIV-1 enhances Mtb growth and spread, the mechanisms by which Mtb exacerbates HIV-1 infection require further investigation (*Bell and Noursadeghi, 2018*; *Deffur et al., 2013*; *Diedrich and Flynn, 2011*).

Besides CD4$^+$ T cells, macrophages are infected by HIV-1 in humans and by the simian immunodeficiency virus (SIV), the most closely related lentivirus to HIV, in non-human primates (NHP) (*Cribbs et al., 2015*; *Rodrigues et al., 2017*). Recently, using a humanized mouse model, macrophages were shown to sustain HIV-1 infection and replication, even in the absence of T cells (*Honeycutt et al., 2017*; *Honeycutt et al., 2016*). This is in line with several studies characterizing tissue macrophages, such as alveolar, microglia and gut macrophages, as reservoirs in HIV-1 patients undergoing antiretroviral therapy (*Ganor et al., 2019*; *Jambo et al., 2014*; *Mathews et al., 2019*; *Sattentau and Stevenson, 2016*).

Macrophages are key host cells for Mtb (*O'Garra et al., 2013*; *VanderVen et al., 2016*). We recently reported the importance of macrophages in HIV-1 exacerbation within the TB co-infection context (*Souriant et al., 2019*). Using relevant in vitro and in vivo models, we showed that TB-associated microenvironments activate macrophages towards an M(IL-10) profile, distinguished by a CD16$^+$CD163$^+$MerTK$^+$ phenotype. Acquisition of this phenotype is dependent on the IL-10/STAT3 signaling pathway (*Lastrucci et al., 2015*). M(IL-10) macrophages are highly susceptible not only to Mtb infection (*Lastrucci et al., 2015*), but also to HIV-1 infection and spread (*Souriant et al., 2019*). At the functional level, we demonstrated that TB-associated microenvironments stimulate the formation of tunneling nanotubes (TNT), membranous channels connecting two or more cells over short to long distances above substrate. TNT are subdivided in two classes based on their thickness and cytoskeleton composition: 'thin' TNT (<0.7 µm in diameter) containing F-actin, and 'thick' TNT (>0.7 µm in diameter) are enriched in F-actin and microtubules (MT) (*Souriant et al., 2019*). Thick TNT are functionally distinguished by the transfer of large organelles, such as lysosomes and mitochondria (*Dupont et al., 2018*; *Eugenin et al., 2009*; *Hashimoto et al., 2016*). While the contribution for each TNT class to HIV-1 pathogenesis has not been explored (*Dupont et al., 2018*; *Eugenin et al., 2009*; *Hashimoto et al., 2016*), we reported that total inhibition of TNT formation in M(IL-10) macrophages resulted in the abrogation of HIV-1 exacerbation induced by Mtb (*Souriant et al., 2019*). Factors influencing TNT function in M(IL-10) macrophages remain unknown at large.

In this study, global mapping of the M(IL-10) macrophage transcriptome revealed Siglec-1 (CD169, or sialoadhesin) as a potential factor responsible for HIV-1 dissemination in the co-infection context with TB. As a type-I transmembrane lectin receptor, Siglec-1 possesses a large extracellular domain composed of 17 immunoglobulin-like domains, including the N-terminal V-set domain, which allows the *trans* recognition of terminal α2,3-linked sialic acid residues in *O*- and *N*-linked glycans and glycolipids, such as those surface-exposed in HIV-1 and SIV particles (*Izquierdo-Useros et al., 2012a*; *Puryear et al., 2012*). While Siglec-1 has yet to be implicated in the TB context, it is clearly involved in the pathogenesis of HIV-1, SIV and other retroviruses (*Martinez-Picado et al., 2017*). Siglec-1 is mainly expressed in myeloid cells (*e.g.* macrophages and dendritic cells) and participates in HIV-1 transfer from myeloid cells to T cells, as well as in the initiation of virus-containing compartment (VCC) formation in macrophages (*Izquierdo-Useros et al., 2012a*; *Izquierdo-Useros et al., 2012b*; *Puryear et al., 2013*; *Puryear et al., 2012*), and in the viral dissemination in vivo (*Akiyama et al., 2017*; *Izquierdo-Useros et al., 2012a*; *Sewald et al., 2015*). Indeed, HIV-1 and other retroviruses have evolved the capacity to hijack the immune surveillance and housekeeping immunoregulatory functions of Siglec-1 (*Izquierdo-Useros et al., 2014*; *O'Neill et al., 2013*). Here, we investigate how Siglec-1 expression is induced by TB, and the role it has in the capture and transfer of HIV-1 by TB-induced M(IL-10) macrophages, in particular in the context of TNT.

## Results

### Tuberculosis-associated microenvironments induce Siglec-1 in macrophages

TB-induced M(IL-10) macrophages are highly susceptible to HIV-1 infection and spread (*Souriant et al., 2019*). To assess the global gene expression landscape in these cells, we performed a genome-wide transcriptome analysis (GEO submission GSE139511). To this end, we employed our

published in vitro model (*Lastrucci et al., 2015*), which relies on the use of conditioned medium from either mock- (cmCTR) or Mtb-infected (cmMTB) human macrophages. As we described and observed before and herein, cmMTB-treated cells were positive for the M(IL-10) markers (CD16+-CD163+MerTK+ and phosphorylated STAT3), and displayed a high rate of HIV-1 infection, as compared to those treated with cmCTR (*Lastrucci et al., 2015*). A distinct 60 gene-transcript signature was defined in cmMTB-treated cells, using a combination of the expression level, statistical filters and hierarchical clustering; 51 genes were up-regulated and nine genes were down-regulated in cmMTB- compared to cmCTR-treated cells (*Figure 1A*). We compared expression data of cmMTB- and cmCTR-treated cells to public genesets available from MSigDB (Broad Institute) using the gene set enrichment analysis (GSEA) algorithm (*Subramanian et al., 2005*). As shown in *Figure 1—figure supplement 1A*, a significant fraction of genes that were up-regulated in response to interferon (IFN) type I (*e.g.* IFNα) and II (*i.e.* IFNγ), were also found, as a group, significantly up-regulated in cmMTB-treated cells in comparison to control samples (FDR q-value:$<10^{-3}$). IFN-stimulated genes (ISG) usually exert antiviral activities (*McNab et al., 2015*; *Schneider et al., 2014*) and cannot be inferred as obvious candidates to facilitate HIV-1 infection. However, among this ISG signature, the up-regulation of Siglec-1 (7.4-fold, adjusted p-value of 0.0162) in cmMTB-treated cells captured our attention due to its known role in HIV-1 pathogenesis (*Izquierdo-Useros et al., 2014*; *O'Neill et al., 2013*). We confirmed a high Siglec-1 expression in cmMTB-treated macrophages at the mRNA (*Figure 1B*), intracellular and cell-surface protein (*Figure 1C* and *Figure 1—figure supplement 1B–D*) levels. This effect was superior to the level obtained in HIV-1-infected cells (*Figure 1—figure supplement 1E*). Particularly, cmMTB-treated macrophages displayed high density of Siglec-1 surface expression applying a quantitative FACS assay that determines the absolute number of Siglec-1 antibody binding sites per cell (*Figure 1D*).

These data indicate that Siglec-1 is highly expressed in human macrophages exposed to TB-associated microenvironments and potentially in the context of TB-HIV co-infection.

## Siglec-1+ alveolar macrophage abundance correlates with pathology in co-infected primates

NHP has been an invaluable in vivo model to better understand the role of macrophages in SIV/HIV pathogenesis (*Merino et al., 2017*). Considering Siglec-1 binds sialylated lipids present in the envelop of HIV-1 and SIV (*Izquierdo-Useros et al., 2012a*; *Puryear et al., 2012*), we examined the presence of Siglec-1 positive alveolar macrophages in lung biopsies obtained from different NHP groups: (i) co-infected with Mtb (active or latent TB) and SIV, (ii) mono-infected with Mtb (active or latent TB), (iii) mono-infected with SIV, and (iv) healthy (*Supplementary file 1*-Table S1, *Figure 1—figure supplement 2A*; *Cai et al., 2015*; *Kuroda et al., 2018*; *Souriant et al., 2019*). Histological immuno-staining confirmed the presence of Siglec-1+ alveolar macrophages in the lungs of healthy NHP (*Figure 1E–F* and *Figure 1—figure supplement 2A*), and revealed its significant increase in NHP mono-infected with either Mtb or SIV (*Figure 1E–F* and *Figure 1—figure supplement 2A*). Strikingly, we noticed a massive abundance of these cells in co-infected NHP (*Figure 1E–F* and *Figure 1—figure supplement 2A*). Concerning the overall abundance of Siglec-1+ leukocytes in lungs, we observed a significant increase in all infected NHP in comparison to healthy, with a higher tendency in active TB or co-infected NHP (*Figure 1G* and *Figure 1—figure supplement 2A*). In fact, the number of Siglec-1+ leukocytes correlated positively with the severity of NHP pathology (*Figure 1H*, *Supplementary file 1*-Table S2). Based on their cell morphology, localization in alveoli, and co-expression with the macrophage marker CD163 (*Figure 1—figure supplement 2A–C*), Siglec-1+ cells were identified as alveolar macrophages.

Collectively, these data show that Siglec-1 is up-regulated in alveolar macrophages in the context of a retroviral co-infection with active TB.

## Siglec-1 expression is dependent on Mtb-induced type I IFN signaling

Siglec-1 is an ISG whose expression is induced by IFN-I in myeloid cells (*Hartnell et al., 2001*). In addition to viral infection, IFN-I is also induced in TB and known to mainly play a detrimental role (*McNab et al., 2015*; *Moreira-Teixeira et al., 2018*). Siglec-1 expression has not been described in the TB context or in co-infection with retroviruses such as SIV or HIV-1, therefore we assessed whether IFN-I stimulates Siglec-1 expression in TB-associated microenvironments. First, we found

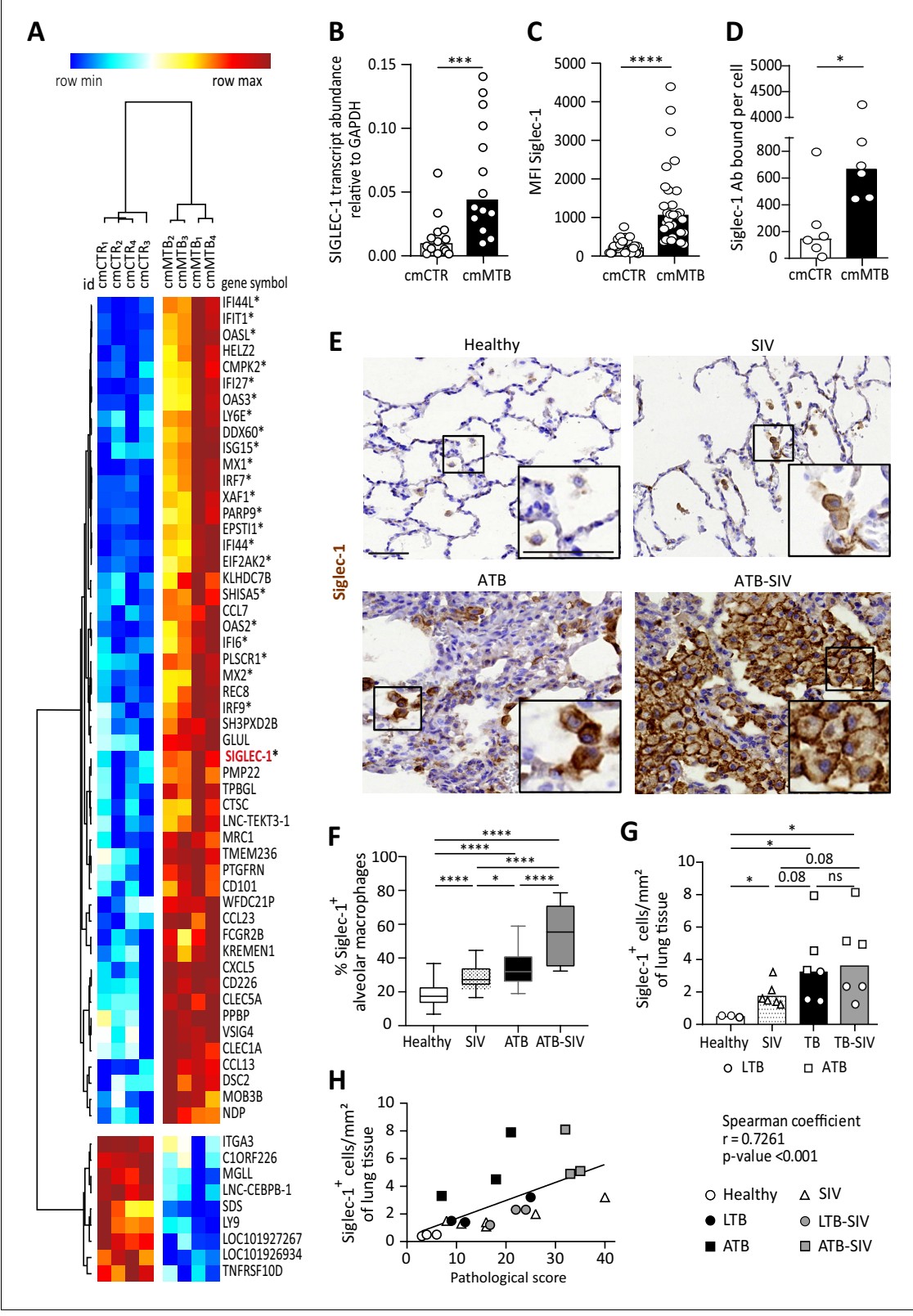

**Figure 1.** Tuberculosis-associated microenvironments induce Siglec-1 expression in macrophages. (A–D) For 3 days, human monocytes were differentiated into macrophages with cmCTR (white) or cmMTB (black) supernatants. (A) Heatmap from a transcriptomic analysis (GEO submission GSE139511) illustrating the top 60 differentially expressed genes (DEGs) between cmCTR- or cmMTB-cells. Selection of the top DEGs was performed using an adjusted p-value ≤ 0.05, a fold change of at least 2, and a minimal expression of 6 in a log₂ scale. Hierarchical clustering was performed using

*Figure 1 continued on next page*

*Figure 1 continued*

the complete linkage method and the Pearson correlation metric with Morpheus (Broad Institute). Interferon-stimulated genes (ISG) are labelled with an asterisk and Siglec-1 is indicated in red. (**B–D**) Validation of Siglec-1 expression in cmMTB-treated macrophages. Vertical scatter plots showing the relative abundance to mRNA (**B**), median fluorescent intensity (MFI) (**C**), and mean number of Siglec-1 antibody binding sites per cell (**D**). Each circle represents a single donor and histograms median values. (**E**) Representative immunohistochemical images of Siglec-1 staining (brown) in lung biopsies of healthy, SIV infected (SIV), active TB (ATB), and co-infected (ATB-SIV) non-human primates (NHP). Scale bar, 100 μm. Insets are 2x zoom. (**F**) Vertical Box and Whisker plot indicating the distribution of the percentage of Siglec-1$^+$ alveolar macrophages in lung biopsies from the indicated NHP groups. Quantification analysis from n = 800 alveolar macrophages grouped from three independent animals per NHP group. (**G**) Vertical scatter plots displaying the number of cells that are positive for Siglec-1 per mm$^2$ of lung biopsies from the indicated NHP groups. Each symbol represents a single animal per NHP group. (**H**) Correlation between Siglec-1$^+$ cells per mm$^2$ of lung tissue and the pathological score for healthy (white circle), SIV$^+$ (white triangles), latent (black circle) or active (black square) TB, and SIV$^+$ with latent (grey circle) or active (grey square) TB. Each symbol represents a single animal per NHP group. Mean value is represented as a black line. Statistical analyses: Two-tailed, Wilcoxon signed-rank test (**B–D**), Mann-Whitney unpaired test (**F–G**), Spearman correlation (**H**). *p<0.05, **p<0.01, ***p<0.001, ****p<0.0001. ns: not significant. See *Figure 1—source data 1*. The online version of this article includes the following source data and figure supplement(s) for figure 1:

**Source data 1.** Raw data and statistical analyses supporting Siglec-1 expression in human and non-human primate macrophages exposed to TB-associated microenvironment.
**Figure supplement 1.** Tuberculosis-associated microenvironments increase Siglec-1 expression in human macrophages.
**Figure supplement 2.** Tuberculosis-associated microenvironments increase Siglec-1 expression in non-human primate alveolar macrophages.

that cmMTB contains high amounts of IFN-I compared to cmCTR (*Figure 2A*). Next, we showed that recombinant IFN-β significantly increased Siglec-1 cell-surface expression in macrophages, close to the level induced by cmMTB (*Figure 2B*). Interestingly, we observed a modest, albeit significant, induction of Siglec-1 expression in cells treated with interleukin 10 (IL-10), a cytokine we have previously shown to be abundant in cmMTB (*Lastrucci et al., 2015*) and that renders macrophages highly susceptible to HIV-1 infection (*Souriant et al., 2019*). However, IL-10 depletion had no effect on Siglec-1 expression by cmMTB-treated cells (*Figure 2C*). By contrast, blocking the IFN-I receptor (IFNAR-2) during cmMTB treatment fully abolished the expression of Siglec-1 (*Figure 2D* and *Figure 1—figure supplement 1D*), indicating that IFN-I is the responsible factor for Siglec-1 up-regulation in cmMTB-treated cells.

IFN-I binding to IFNAR leads to the phosphorylation and nuclear translocation of the transcription factor STAT1, whose role is essential for transcription of ISG (*Ivashkiv and Donlin, 2014*). We thus examined the status of STAT1 activation in co-infected NHP lung tissue. Histological staining of serial sections of co-infected lungs revealed that zones rich in Siglec-1$^+$ leukocytes also exhibited positivity for nuclear phosphorylated STAT1 (pSTAT1) (*Figure 2E*), and the abundance of these two markers strongly correlated with the active status of TB in the different NHP groups (*Figure 2F*). Moreover, we found that the majority of Siglec-1$^+$ alveolar macrophages were also positive for nuclear pSTAT1 in the infected NHP groups compared to healthy (*Figure 2E and G*). In fact, there was a higher number of pSTAT1$^+$ alveolar macrophages in TB-SIV co-infected lungs when compared to those from mono-infected NHP and this number directly correlates with the number of Siglec-1$^+$ alveolar macrophages (*Figure 2G–H*).

Altogether, these data demonstrate that Siglec-1 expression in human macrophages is controlled by IFN-I in a TB-associated microenvironment, and suggest the involvement of the IFN-I/STAT1/Siglec-1 axis in the pathogenesis of TB and co-infection with retroviruses.

## Siglec-1 localization on thick TNT is associated with their length and HIV-1 cargo

TNT formation is responsible for the increase in HIV-1 spread between human macrophages in TB-associated microenvironments (*Souriant et al., 2019*). To investigate whether Siglec-1 expression is involved in this process, we first examined its localization in the context of TNT formed by cmMTB-treated cells infected by HIV-1. We observed that Siglec-1 is localized mainly on microtubule (MT)-positive thick TNT, and not on thin TNT (*Figure 3A* and *Figure 3—video 1*). Semi-automatic quantification of hundreds of TNT showed that about 50% of thick TNT were positive for Siglec-1 (*Figure 3B* and *Figure 3—figure supplement 1A*). These TNT exhibited a greater length compared to those lacking Siglec-1 (*Figure 3C*). Importantly, unlike thin TNT, HIV-1 viral proteins are found mainly inside Siglec-1$^+$ thick TNT (*Figure 3D–E* and *Figure 3—video 2*). In addition, these thick TNT

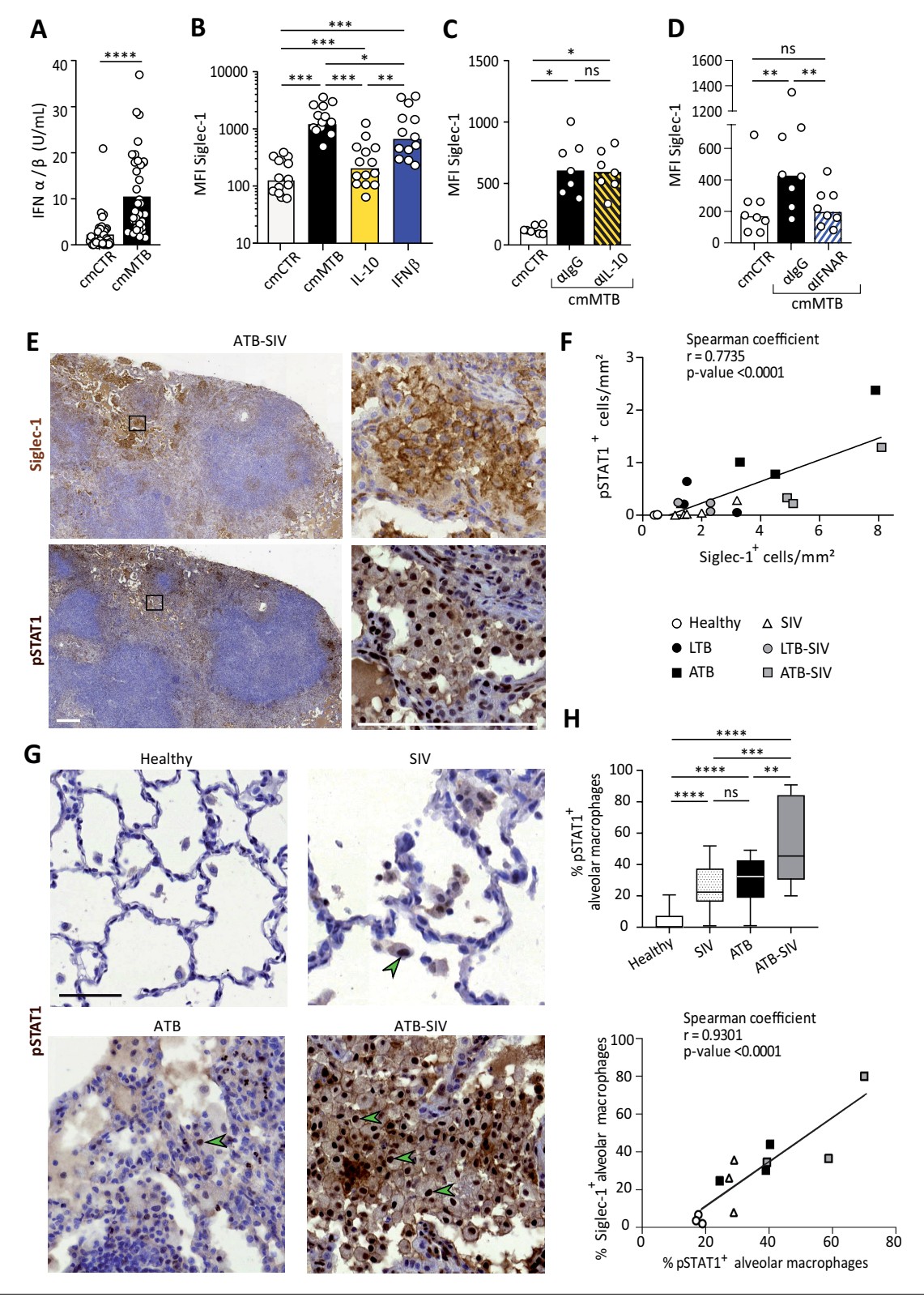

**Figure 2.** Siglec-1 expression is dependent on Mtb-induced type I IFN signaling. (**A**) Vertical scatter plot showing the relative abundance of IFN-I in cmCTR (white) and cmMTB (black) media, as measured indirectly after 24 hr exposure to the HEK-Blue IFN-α/β reporter cell line yielding reporter activity in units (U) per mL. (**B–D**) Vertical scatter plots displaying the median fluorescent intensity (MFI) of Siglec-1 cell-surface expression after three days of monocyte differentiation into macrophages either with cmMTB (black) or cmCTR (white), the indicated recombinant cytokines (**B**), the presence

*Figure 2 continued on next page*

*Figure 2 continued*

of an IL-10 depletion (α-IL-10) or a control (α-IgG) antibodies (**C**), or the presence of an IFNAR-2 blocking (α-IFNAR) or control (α-IgG) antibodies (**D**). (**E**) Representative serial immunohistochemical images of lung biopsies of a co-infected (ATB-SIV) NHP stained for Siglec-1 (brown, top) and pSTAT1 (brown, bottom). Scale bar, 250 μm. Insets are 10x zooms. (**F**) Correlation of the percentage of cells positive for Siglec-1 and pSTAT1, as measured per mm$^2$ of lung tissue from the indicated NHP groups. Mean value is represented as a black line. (**G**) Representative immunohistochemical images of lung biopsies from the indicated NHP group stained for pSTAT1 (brown). Arrowheads show pSTAT1-positive nuclei. Scale bar, 500 μm. (**H**) Upper panel: Vertical Box and Whisker plot illustrating the percentage of pSTAT1$^+$ alveolar macrophages in lung biopsies from the indicated NHP groups. Quantification analysis from n = 600 alveolar macrophages grouped from three independent animals per NHP group. Lower panel: Correlation of the percentage of alveolar macrophages positive for Siglec-1 and pSTAT1, from the indicated NHP groups. Mean value is represented as a black line. (**A–D**) Each circle within vertical scatter plots represents a single donor and histograms median value. Statistical analyses: Two-tailed, Wilcoxon signed-rank test (**A–D**), Spearman correlation (**F**, **H** lower panel), and Mann-Whitney unpaired test (**H**, upper panel). *p<0.05, **p<0.01, ***p<0.001, ****p<0.0001. ns: not significant. See *Figure 2—source data 1*.

The online version of this article includes the following source data for figure 2:

**Source data 1.** Raw data and statistical analyses supporting that IFN-I induced by M. tuberculosis is responsible for Siglec-1 expression in human and non-human primate macrophages.

---

also contained large organelles such as mitochondria (*Figure 3F* and *Figure 3—figure supplement 1B*), another characteristic distinguishing thick from thin TNT (*Dupont et al., 2018*; *Onfelt et al., 2006*). In general, we also noticed that the incidence of Siglec-1$^+$ thick TNT between HIV-1 infected macrophages persisted for more than one week upon HIV-1 infection (*Figure 3—figure supplement 1C*), suggesting a high degree of stability for these TNT.

These findings reveal a strong localization of Siglec-1 on MT-positive thick TNT that correlates positively with a greater length and high cargo of HIV-1 or mitochondria, arguing for a functional capacity of Siglec-1$^+$ TNT to transfer material to recipient cells over long distances.

## The Mtb-induced exacerbation of HIV-1 infection and spread in macrophages requires Siglec-1

To evaluate a functional role for Siglec-1 in the susceptibility of macrophages to HIV-1 infection and spread induced by TB, Siglec-1 was depleted in cmMTB-treated cells by siRNA-mediated gene silencing (*Figure 4A* and *Figure 4—figure supplement 1A*). While this depletion did not affect the total number of thick TNT (*Figure 4B* and *Figure 4—figure supplement 1B*), we observed a 2-fold shortening of thick TNT in cells lacking Siglec-1 when compared to control cells (*Figure 4C*). Then, we performed a viral uptake assay in these cells using HIV-1-Gag-eGFP virus-like particles (GFP VLP) lacking the viral envelope glycoprotein but bearing sialylated lipids that interact with Siglec-1 on myeloid cells (*Izquierdo-Useros et al., 2012b*; *Puryear et al., 2013*). We consistently observed binding of VLP along Siglec-1$^+$ thick TNT (*Figure 4—figure supplement 1C*). Yet, in the absence of Siglec-1, we noticed a significant reduction of VLP binding in comparison to control cells (*Figure 4—figure supplement 1D*). We confirmed this functional observation using a blocking monoclonal antibody against Siglec-1, showing that this receptor is involved in HIV-1 binding in cmMTB-treated cells (*Figure 4—figure supplement 1E*).

We then assessed the role of Siglec-1 in HIV-1 transfer between macrophages, as this receptor is also important for the transfer of the virus to CD4$^+$ T cells (*Akiyama et al., 2015*; *Izquierdo-Useros et al., 2012a*; *Puryear et al., 2013*). We used an established co-culture system between cmMTB-treated macrophages that allows the transfer of the viral Gag protein from infected (donor, Gag$^+$, red) to uninfected (recipient, CellTracker$^+$, green) cells over 24 hr (*Souriant et al., 2019*; *Figure 4—figure supplement 1F*). Of note, since Siglec-1 facilitates the infection of macrophages (*Zou et al., 2011*), we used VSV-G pseudotyped viruses to avoid any effect on HIV-1 primo-infection. Like this, we ensured the viral content was equal in cells at the time of the co-culture despite the loss of Siglec-1 (*Figure 4—figure supplement 1G*). The siRNA-mediated depletion of Siglec-1 significantly diminished the capacity of cmMTB-treated macrophages to transfer HIV-1 to recipient cells (*Figure 4D*), indicating that this receptor is involved in the macrophage-to-macrophage viral spread (*Souriant et al., 2019*). Intriguingly, there was a decreasing tendency for the capacity of Siglec-1-depleted cmMTB-treated macrophages to transfer mitochondria to recipient cells compared to controls (*Figure 4E* and *Figure 4—figure supplement 1F*), alluding to a possible defect in mechanisms involved in intercellular material transfer including through thick TNT (*Torralba et al., 2016*).

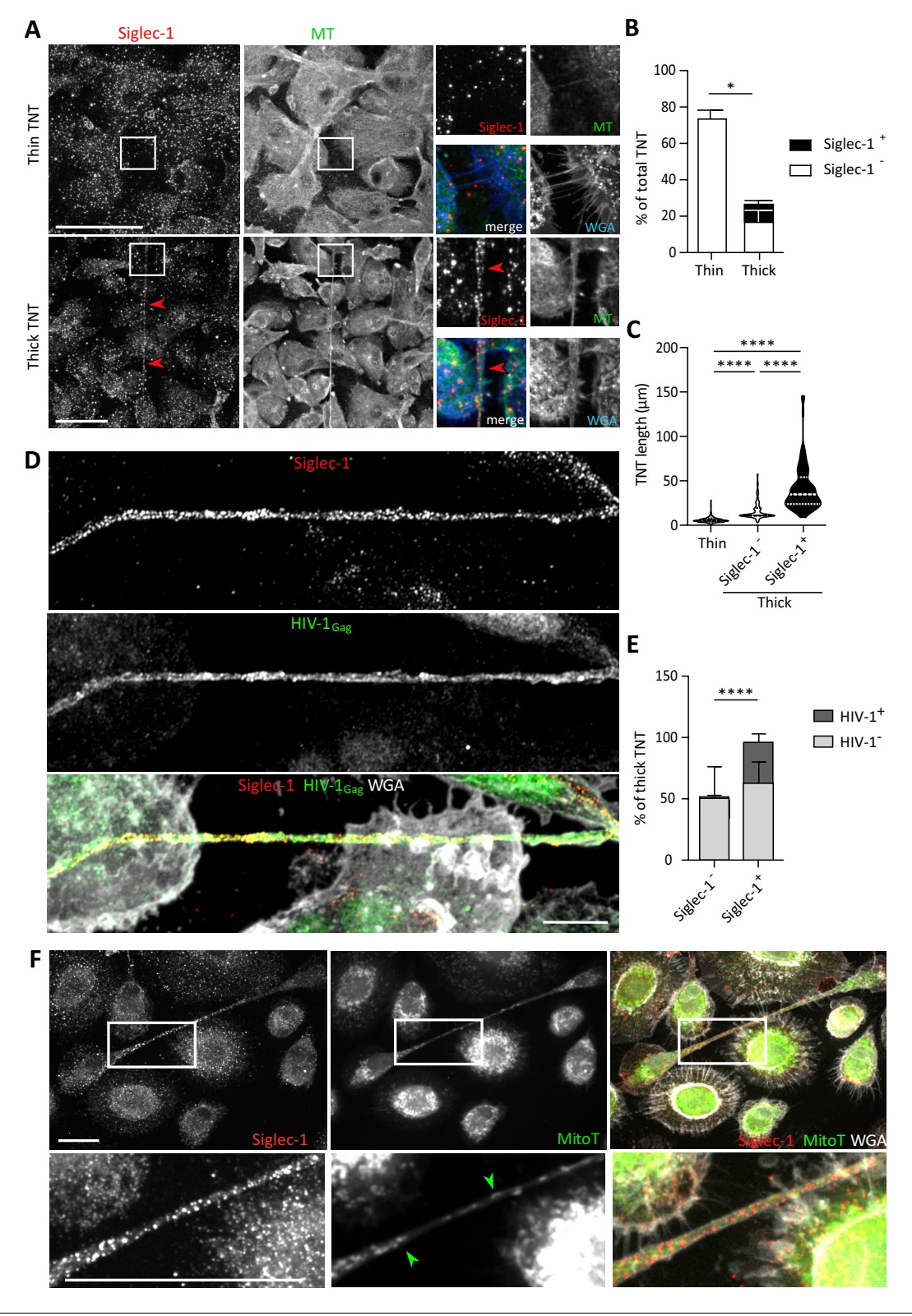

**Figure 3.** Siglec-1 localization on thick TNT is associated with their length and HIV-1/mitochondria cargo. (A–F) Human monocytes were differentiated into macrophages with cmMTB for 3 days, and then infected with HIV-1-ADA strain (unless indicated otherwise) and fixed 3 days post-infection. (A) Representative immunofluorescence images of cmMTB-treated macrophages infected with HIV-1-ADA, and stained for extracellular Siglec-1 (red), intracellular tubulin (MT, green) and Wheat Germ Agglutinin (WGA, blue). Inserts are 3x zooms. Red arrowheads show Siglec-1 localization on TNT.
*Figure 3 continued on next page*

*Figure 3 continued*

Scale bar, 20 µm. (B) Vertical bar plot showing the semi-automatic quantification of Siglec-1$^+$ TNT (black) and Siglec-1$^-$ TNT (white) in thick (WGA$^+$, MT$^+$) and thin (WGA$^+$, MT$^-$) TNT. 400 TNT were analyzed from two independent donors. (C) Siglec-1$^+$ TNT exhibit a larger length index. Violin plots displaying the semi-automatic quantification of TNT length (in µm) for thin (WGA$^+$, MT$^-$), and thick TNT (WGA$^+$, MT$^+$) expressing Siglec-1 or not. 400 TNT were analyzed per condition from two independent donors. (D) Representative immunofluorescence images of cmMTB-treated macrophages 3 day post-infection with HIV-1-NLAD8-VSVG strain, and stained for extracellular Siglec-1 (red), intracellular HIV-1$_{Gag}$ (green) and WGA (grey). Scale bar, 10 µm. (E) Vertical bar plots indicating the quantification of presence (dark grey) or absence (light grey) of HIV-1$_{Gag}$ in thick TNT (WGA$^+$, MT$^+$) expressing Siglec-1 or not. 120 TNT in at least 1000 cells were analyzed from four independent donors. (F) Representative immunofluorescence images of cmMTB-treated macrophages infected with HIV-1-ADA loaded with MitoTracker (MitoT, green), and stained for extracellular Siglec-1 (red) and WGA (grey). Inserts are 3x zooms. Green arrowheads show mitochondria inside TNT. Scale bar, 10 µm. Statistical analyses: Two-way ANOVA comparing the presence of Siglec-1 in thin and thick TNT (B), and two-tailed Mann-Whitney unpaired test comparing TNT length (C) and the presence of HIV-1 in TNT (E). *p<0.05, ****p<0.0001. See *Figure 3—source data 1*.

The online version of this article includes the following video, source data, and figure supplement(s) for figure 3:

**Source data 1.** Raw data and statistical analyses supporting Siglec-1 expression on thick TNT and its correlation with TNT length.

**Figure supplement 1.** Siglec-1 localizes specifically on thick tunneling nanotubes that contain HIV-1$_{Gag}$ and mitochondria.

**Figure 3—video 1.** Related to *Figure 3A*.

https://elifesciences.org/articles/52535#fig3video1

**Figure 3—video 2.** Related to *Figure 3D*.

https://elifesciences.org/articles/52535#fig3video2

Remarkably, using replicative HIV-1 ADA strain (*Figure 4A*), we showed that silencing Siglec-1 expression in cmMTB-treated cells abolished the exacerbation of HIV-1 infection and production, as well as the enhanced formation of multinucleated giant cells (MGC) (*Figure 4F–G*), which are pathological hallmarks of HIV-1 infection of macrophages (*Vérollet et al., 2015*; *Vérollet et al., 2010*).

These results determine that TB-induced Siglec-1 expression plays a key part in HIV-1 uptake and efficient cell-to-cell transfer, resulting in the exacerbation of HIV-1 infection and production in M(IL-10) macrophages.

## Discussion

In this study, we investigated potential mechanisms by which Mtb exacerbates HIV-1 infection in macrophages, and uncovered a deleterious role for Siglec-1 in this process. These findings have different contributions to our understanding of this receptor in the synergy between Mtb and distinct retroviral infections, and also for TNT biology in host-pathogen interactions.

Our global transcriptomic approach revealed the up-regulation of Siglec-1, as part of an ISG-signature enhanced in macrophages exposed to a TB-associated microenvironment. Although pulmonary active TB has been characterized as an IFN-I-driven disease (*Berry et al., 2010*; *McNab et al., 2015*; *Moreira-Teixeira et al., 2018*), there are no report in the literature about a role for Siglec-1 in TB or in Mtb co-infection with retroviruses. Expression of Siglec-1 is restricted to myeloid cells except circulating monocytes (*Crocker et al., 2007*), and is enhanced by IFN-I (*Puryear et al., 2013*; *Rempel et al., 2008*) and during HIV-1 infection (*Pino et al., 2015*). In addition, human alveolar macrophages are distinguished from lung interstitial macrophages by Siglec-1 expression (*Yu et al., 2016*). In this study, we determined that IFN-I present in TB-associated environment is responsible for Siglec-1 overexpression in human macrophages, which resembled that obtained in HIV-1-infected cells. While we saw a modest induction of Siglec-1 in macrophages upon IL-10 treatment, its depletion from the TB-associated microenvironment had no effect on Siglec-1 expression. This could be explained by the fact that IL-10 induces the autocrine production of IFN-I (*Ziegler-Heitbrock et al., 2003*) to indirectly modulate Siglec-1 expression in M(IL-10) macrophages, which then contributes to the exacerbation of HIV-1 infection as we previously reported (*Souriant et al., 2019*). In the context of the most closely related lentivirus to HIV, namely SIV, we not only confirmed the presence of Siglec-1$^+$ alveolar macrophages in SIV-infected NHP, but also reported the high abundance of these cells in active TB and in co-infected NHP groups, when compared to healthy ones. Importantly, we associated the high abundance of Siglec-1$^+$ leukocytes with the increase NHP pathological scores, and it correlated positively to the detection of pSTAT1$^+$ macrophage nuclei in histological staining of serial sections of lung biopsies from co-infected NHP. This is in line with a recent report on the

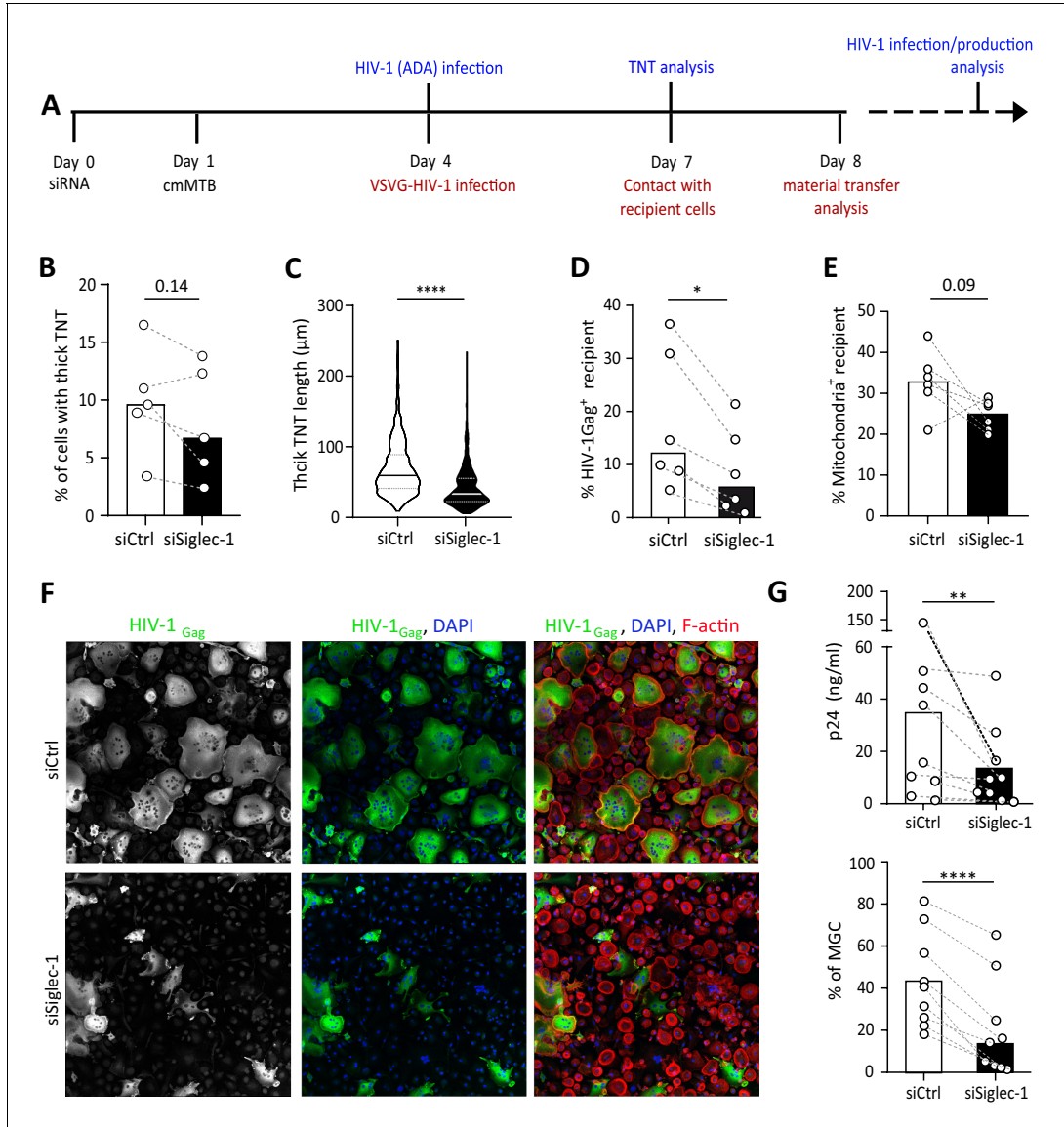

**Figure 4.** The exacerbation of HIV-1 infection and spread in macrophages treated with cmMTB requires Siglec-1. (**A**) Experimental design. Monocytes from healthy subjects were transfected with siRNA targeting of Siglec-1 (siSiglec-1, black) or not (siCtrl, white). A day after, monocytes were differentiated into macrophages with cmMTB for 3 days. Cells were then infected with HIV-1-ADA (blue protocol) to measure the formation (**B**) and length (**C**) of TNT at day 7, or assess HIV-1 production and multinucleated giant cell (MGC) formation at day 14 (**F–G**). In parallel, cells were either infected with HIV-NLAD8-VSVG or labelled with mitoTracker to measure the transfer (red protocol) of HIV-1 (**D**) or mitochondria (**E**), respectively. (**B**) Before-and-after plots showing the percentage of cells forming thick TNT (F-actin$^+$, WGA$^+$, MT$^+$). (**C**) Violin plots displaying the semi-automatic quantification of TNT length (in µm) for thick (WGA$^+$, MT$^+$) TNT; 300 TNT were analyzed per condition from two independent donors. (**D–E**) Before-and-after plots indicating the percentage of HIV-1$_{Gag}^+$ cells (**D**) or MitoTracker$^+$ cells (**E**) among CellTracker$^+$ cells after 24 hr co-culture. (**F**) Representative immunofluorescence images of siRNA transfected cells treated with cmMTB, 14 days post-HIV-1 infection. Cells were stained for intracellular HIV-1$_{Gag}$ (green), F-actin (red) and DAPI (blue). Scale bar, 500 µm. (**G**) Vertical scatter plots showing HIV-1-p24 concentration in cell supernatants (upper panel) and percentage of MGC (lower panel) at day 14 post-HIV-1 infection in cells represented in F (siSiglec-1, black; siCtrl, white). (**B, D, E and G**) Each circle represents a single donor and histograms median value. Statistical analyses: Paired t-test (B, G lower panel) or two-tailed, Wilcoxon signed-rank test (C-E, G upper panel). *p<0.05, **p<0.01, ****p<0.0001. See *Figure 4—source data 1*.

The online version of this article includes the following source data and figure supplement(s) for figure 4:

**Source data 1.** Raw data and statistical analyses supporting a role for Siglec-1 in TNT length, HIV-1 and mitochondrial cell-to-cell trasfer, and exacerbation of HIV-1 infection.

**Figure supplement 1.** Siglec-1 is required for the capture and transfer of HIV-1 in cmMTB-treated macrophages.

**Figure supplement 1—source data 1.** Supplemental raw data and statistical analyses supporting the functional role of Siglec-1 in human macrophages using an siRNA-mediated gene silencing approach.

presence of IFN-I, IFNAR and different ISG in alveolar and lung interstitial tissue from NHP with active TB (**Mattila, 2019**), and with the fact that the in vivo expression of Siglec-1 is up-regulated early in myeloid cells after SIV infection and maintained thereafter in the pathogenic NHP model (**Jaroenpool et al., 2007**). In TB-SIV co-infection, we hypothesized that IFN-I is not exerting the expected antiviral effect, but instead is concomitant with chronic immune activation and attenuated by the high expression of Siglec-1 in myeloid cells, as recently proposed in the HIV-1 context (**Akiyama et al., 2017**). Altogether, these findings uncover the IFN-I/STAT1/Siglec-1 axis as a mechanism established by Mtb to exacerbate HIV-1 infection in myeloid cells, and call for the need to further investigate this signaling pathway in TB pathogenesis.

Another aspect worth highlighting is the impact that Siglec-1 expression has in the capture and transfer of HIV-1 by M(IL-10) macrophages, in particular in the context of TNT. First, we reported that Siglec-1 is located on MT-positive thick (and not on thin) TNT, correlating positively with increased length and HIV-1 cargo. To our knowledge, no receptor has been described so far to be present mainly on thick TNT, making Siglec-1 an unprecedented potential marker for this subtype of TNT (**Dupont et al., 2018**). Second, viral uptake assays demonstrated the functional capacity of Siglec-1, including on thick TNT, to interact with viral-like particles bearing sialylated lipids; loss-of-function approaches showed Siglec-1 is important in the capture of these viral particles. Third, Siglec-1 depletion correlated with a decrease in thick TNT length, but had no effect in the total number of thick TNT. This suggests that, while the IFN-I/STAT1 axis is responsible for Siglec-1 expression in M(IL-10) macrophages, it does not contribute to TNT formation. This is line with our previous report where TNT formation induced by TB-associated microenvironments depended on the IL-10/STAT3 axis (**Souriant et al., 2019**). Concerning the shortening of thick TNT length, we infer that it may reflect a fragile state due to an altered cell membrane composition in the absence of Siglec-1; TNT are known for their fragility towards light exposure, shearing force and chemical fixation (**Rustom et al., 2004**). We hypothesize that the longer the TNT is, the more rigidity it requires to be stabilized. Cholesterol and lipids are known to increase membrane rigidity (**Redondo-Morata et al., 2012**) and are thought to be critical for TNT stability (**Lokar et al., 2012**; **Thayanithy et al., 2014**). Thus, the presence of Siglec-1 in thick TNT may affect the cholesterol and lipid composition *via* the recruitment of GM1/GM3 glycosphingolipid-enriched microvesicles (**Halász et al., 2018**). In fact, TNT formation depends on GM1/GM3 ganglioside and cholesterol content (**Kabaso et al., 2011**; **Lokar et al., 2012**; **Osteikoetxea-Molnár et al., 2016**; **Tóth et al., 2017**). Since GM1 and GM3 glycosphingolipids are *bona fide* ligands for Siglec-1 (**Puryear et al., 2013**), it is likely that Siglec-1[+] thick TNT exhibit a higher lipid and cholesterol content, and hence an increase of membrane rigidity that favors the stability of longer TNT. Fourth, Siglec-1-depleted donor macrophages were less capable to transfer HIV-1, and to some extend mitochondria, to recipient cells. While infectious synapse and exososome release are mechanisms attributed to Siglec-1 that contribute to cell-to-cell transfer of HIV-1 (**Bracq et al., 2018**; **Gummuluru et al., 2014**; **Izquierdo-Useros et al., 2014**), they accomplish so extracellularly. Here, we speculate that Siglec-1 participates indirectly in the intracellular HIV-1 transfer *via* TNT as a tunnel over long distance, suggesting that factors affecting TNT rigidity favor distal viral dissemination while ensuring protection against immune detection. Independent of HIV-1 infection, we also noticed that cmMTB-treated cells depleted for Siglec-1 displayed a decreasing tendency to transfer mitochondria among them. As TNT-transferred mitochondria are known to alter the metabolism and functional properties of recipient cells under steady state conditions or in the cancer context, it implies that Siglec-1 may also influence key metabolic pathways such as glycolysis, pentose phosphate and lipid metabolism, among others (**Hekmatshoar et al., 2018**). This is important because, for example, the gain of cancer drug resistance is directly associated to TNT-mediated mitochondria transfer, thus Siglec-1 may represent a novel therapeutic strategy to overcome cancer cell drug resistance. Finally, the depletion of Siglec-1 abrogated the exacerbation of HIV-1 infection and production induced by TB in M(IL-10) macrophages. This is likely to result from an accumulative effect of deficient capture and transfer of HIV-1 in the absence of Siglec-1. However, these results do not discern the specific contribution of Siglec-1 to the cell-to-cell transmission of HIV-1 *via* TNT from that obtained through other mechanisms (**Bracq et al., 2018**). Future studies will address whether the contribution of Siglec-1 to cell-to-cell transfer mechanisms has an impact in Mtb dissemination (**Onfelt et al., 2006**).

In conclusion, our study identifies Siglec-1 as a key factor involved in the exacerbation of HIV-1 infection in macrophages conditioned by cmMTB. It is worth noting that we have previously

reported that a loss-of-function variant in Siglec-1 in the human population does not conclusively establish a role for this receptor in AIDS progression, even though ex vivo experiments demonstrated that cells from these individuals were functionally null or partially defective for Siglec-1 expression along with poor HIV-1 capture and transmission (*Martinez-Picado et al., 2016*). While this may be counterintuitive for proposing Siglec-1 as a therapeutic target to limit viral dissemination in the co-infection context, there are several challenges to the study of Siglec-1 variants, such as limited cohort size, the lack of complete clinical records, and the restriction to study only off-therapy periods, among others (*Martinez-Picado et al., 2017*). Therefore, there is a strong need for future work targeting Siglec-1 to unveil its in vivo contribution of the mononuclear phagocyte system to HIV-1 pathogenesis to fully exploit the therapeutic potential of this receptor. Beyond the infectious disease context, our study also sheds light on a new homeostatic function for Siglec-1 in human macrophages, such as intercellular communication facilitated by TNT. We argue that Siglec-1 localization on thick TNT has a physiological significance to macrophage biology in health and disease.

# Materials and methods

## Key resources table

| Reagent type (species) or resource | Designation | Source or reference | Identifiers | Additional information |
|---|---|---|---|---|
| *M. tuberculosis* | H37Rv | *Derived from E.R. Baldwin's human-lung isolate H37 by W. Steenken New York, United States, 1934* | (ATCC 25618) | |
| HIV-1 | ADA | Gift from Dr. S Benichou Institut Cochin, Paris, France | N/A | |
| HIV-1 | NLAD8 | Gift from Dr. S Benichou Institut Cochin, Paris, France | N/A | |
| HIV-1 | ADA Gag-iGFP-VSVG | This paper | This paper | |
| Buffy Coat | Leukocytes | Etablissement Français du Sang, Toulouse, France | N/A | |
| Lung biopsies from rhesus macaques | Histological slides | Tulane National Primate Research Center | N/A | |
| Cell line (human HeLa JC.53) | TZM-bl | NIH AIDS Reagent Program | Cat# 8129 | Cultured in: DMEM, 90%; FBS, 10%; 100 units of Penicillin and 0.1 mg/mL of Streptomycin |
| Cell line (human) | HEK-293T | NIH AIDS Reagent Program | Cat# 3318 | Cultured in: DMEM, 10% FCS |
| Cell line (human) | HEK-Blue IFN-α/β Cells | Invivogen | Cat# hkb-ifnab | Cultured in: DMEM, 4.5 g/l glucose, 2 mM L-glutamine, 10% (v/v) heat-inactivated fetal bovine serum, 100 U/ml penicillin, 100 µg/mL streptomycin, 100 µg/mL Normocin |
| Transfected construct (human) | siRNA to Siglec-1 (SMART-Pool) | Horizon Discovery | Cat# L-017521-01-0020 | (200 nM) |
| Transfected construct (human) | siRNA scramble (SMART Pool) | Horizon Discovery | Cat# D-001810-10-50 | (200 nM) |
| Antibody | Mouse monoclonal antihuman Siglec-1 (clone 7-293) | Biolegend | Cat# 346008 RRID:AB_11147948 | FACS (1 µg/mL) |
| Antibody | Mouse monoclonal antihuman CD16 (clone 3G8) | Biolegend | Cat# 302019 and 302018; RRID:AB_492974 and AB_314218 | FACS (1 µg/mL) |
| Antibody | Mouse monoclonal antihuman CD163 (clone GHI/61) | Biolegend | Cat# 333608 RRID:AB_2228986 | FACS (1 µg/mL) |

*Continued on next page*

Continued

| Reagent type (species) or resource | Designation | Source or reference | Identifiers | Additional information |
|---|---|---|---|---|
| Antibody | Mouse monoclonal antihuman MerTK (clone 590H11G1E3) | Biolegend | Cat# 367607 RRID:AB_2566400 | FACS (1 µg/mL) |
| Antibody | Rabbit monoclonal antihuman STAT1 (clone 42H3) | Cell Signaling Technology | Cat# 9175 RRID:AB_2197984 | WB (1:100) |
| Antibody | Rabbit antihuman actin (a.a. 2033) | SigmaAldrich | Cat# A5060 RRID:AB_476738 | WB (1:100) |
| Antibody | Rabbit polyclonal anti-a-tubulin | Abcam | Cat# ab18251 RRID:AB_2210057 | IF (5 µg/mL) |
| Antibody | Mouse monoclonal anti-Siglec-1 (clone hsn 7D2) | Novus Biologicals | Cat# NB 600-534 RRID:AB_526814 | IF (10 µg/mL) IHC (1:200) |
| Antibody | Mouse monoclonal anti-Gag RD1 (clone KC57) | NIH AIDS Reagent program | Cat# 13449 | IF (1:200) |
| Antibody | Mouse monoclonal anti-HIV-1 p24 (clone 183-H12-5C) | NIH AIDS Reagent Program | Cat# 3537 | ELISA (2.5 µg/mL) |
| Antibody | Human polyclonal anti-HIV Immune Globulin (HIVIG) | NIH AIDS Reagent Program | Cat# 3957 | ELISA (6.25 µg/mL) |
| Antibody | Polyclonal goat anti-human IgG | Sigma-Aldrich | Cat# A0170 | ELISA (1:10000) |
| Antibody | Mouse monoclonal antihuman CD163 (clone 10D6) | Leica/Novocastra | Cat# NCL-L-CD163 RRID:AB_2756375 | IHC (1:100) |
| Antibody | Anti-pSTAT1 | Cell Signaling Technology | Cat# 9167 RRID:AB_561284 | WB (1:100) |
| Antibody | Mouse monoclonal anti-IFNAR2 (clone MMHAR-2) | Thermo Fisher Scientific | Cat# 213851 RRID:AB_223508 | Blocking (20 µg/mL) FACS (1 µg/mL) |
| Antibody | Mouse IgG2a isotype control | Thermo Fisher Scientific | Cat# 02-6200 RRID:AB_2532943 | Blocking (20 µg/mL) IF (0.6 µg/mL) |
| Antibody | Polyclonal F(ab)2 goat antirabbit IgG, AlexaFluor 555 | Thermo Fisher Scientific | Cat# A-21430 RRID:AB_2535851 | IF (2 µg/mL) |
| Antibody | Polyclonal F(ab)2 goat antimouse IgG, AlexaFluor 488 | Thermo Fisher Scientific | Cat# A-10684 RRID:AB_2534064 | IF (2 µg/mL) |
| Antibody | Plyclonal F(ab)2 goat antimouse IgG, AlexaFluor 555 | Cell Signaling Technology | Cat# 4409 RRID:AB_1904022 | IF (2 µg/mL) |
| Antibody | Polyclonal goat antirabbit IgG, HRP | Thermo Fisher Scientific | Cat# 32460 RRID:AB_1185567 | WB (1:10000) |
| Antibody | Polyclonal goat antimouse IgG, HRP | Thermo Fisher Scientific | Cat# 31430 RRID:AB_228307 | WB (1:10000) |
| Cytokine (recombinant, human) | M-CSF | Peprotech | Cat# 30025 | (20 ng/mL) |
| Cytokine (recombinant, human) | IFNb | Peprotech | Cat# 300-02BC | 10 and 100 U/mL |
| Cytokine (recombinant, human) | IL-10 | Peprotech | Cat# 200-10 | 10 ng/mL |
| Monocyte isolation | Mouse antihuman CD14 microbeads | Miltenyi Biotec | Cat# 130050201 | |

*Continued*

| Reagent type (species) or resource | Designation | Source or reference | Identifiers | Additional information |
|---|---|---|---|---|
| Monocyte isolation | LS magnetic columns | Miltenyi Biotec | Cat# 130042401 | |
| Western blot | Amersham ECL Prisme Western Blotting Detection Reagent | GE Healthcare | Cat# RPN2232 | |
| Western blot | SuperSignal WestPico Chemiluminescent Substrate | Thermo Scientific | Cat# 34080 | |
| ELISA | IL10 ELISA set | BD Bioscience | Cat# 555157 | |
| Cell culture | Trypsin EDTA 0.05% | Thermo Fisher Scientific | Cat# 25200072 | |
| Cell culture | Accutase | Sigma-Aldrich | Cat# A-6964 | |
| Probe | Phalloidin AlexaFluor 488 | Thermo Fisher Scientific | Cat# A12379 | (33 mM) |
| Probe | Phalloidin Alexa Fluor 647 | Thermo Fisher Scientific | Cat# A22287 | (33 mM) |
| Probe | DAPI | Sigma Aldrich | Cat# D9542 | (500 ng/mL) |
| Probe | CellTracker Green CMFDA Dye | Thermo Fisher Scientific | Cat# C7025 | (500 ng/mL) |
| Probe | MitoTracker Deep Red FM | Invitrogen | Cat# M22426 | (500 ng/mL) |
| IF | Fluorescence Mounting Medium | Agilent Technologies | Cat# S3023802 | |
| IF | Antibody diluent, Background reducing | DAKO, Agilent Technologies | Cat# S302283-2 | |
| Software | ImageJ | ImageJ | http://www.imagej.nih.gov/ij | |
| Software | Prism (v8.0.0) | GraphPad | http://www.graphpad.com | |
| Software | Photoshop CS3 | Adobe | http://www.adobe.com | |
| Software | Adobe Illustrator CS5 | Adobe | https://www.adobe.com/fr/products/illustrator.html | |
| Software | Huygens Professional Version 16.10 | Scientific Volume Imaging | https://svi.nl/HuygensProfessional | |
| Software | FACS DIVA | BD Bioscience | http://www.bdbiosciences.com/ | |
| Software | FlowJo_v10 | FlowJo | https://www.flowjo.com/ | |
| Software | FCS Express V3 | DeNovo Software | http://www.denovosoftware.com | |
| Software | Image Lab | BioRad Laboratories | http://www.biorad.com | |
| Software | Pannoramic Viewer | 3DHISTECH | https://www.3dhistech.com/pannoramic_viewer | |

## Human subjects

Monocytes from healthy subjects were provided by Etablissement Français du Sang (EFS), Toulouse, France, under contract 21/PLER/TOU/IPBS01/20130042. According to articles L12434 and R124361 of the French Public Health Code, the contract was approved by the French Ministry of Science and Technology (agreement number AC 2009921). Written informed consents were obtained from the donors before sample collection.

## Non-Human primate (NHP) samples

All animal procedures were approved by the Institutional Animal Care and Use Committee of Tulane University, New Orleans, LA and were performed at the Tulane TNPRC, and under approval from IACUC (project numbers P3733, P3794, P3373 and P3628). They were performed in strict accordance with NIH guidelines. The twenty adult rhesus macaques used in this study (*Supplementary file 1*-Table S1 and S2) were bred and housed at the Tulane National Primate

Research Center (TNPRC). All macaques were infected as previously described (*Foreman et al., 2016*; *Mehra et al., 2011*; *Souriant et al., 2019*). Briefly, aerosol infection was performed on macaques using a low dose (25 CFU implanted) of Mtb CDC1551. Nine weeks later, a subgroup of the animals was additionally intravenously injected with 300 TCID50 of SIVmac239 in 1 mL saline, while controls received an equal volume of saline solution. Euthanasia criteria were presentation of four or more of the following conditions: (i) body temperatures consistently greater than 2 °F above pre-infection values for three or more weeks in a row; (ii) 15% or more loss in body weight; (iii) serum CRP values higher than 10 mg/mL for three or more consecutive weeks, CRP being a marker for systemic inflammation that exhibits a high degree of correlation with active TB in macaques (*Kaushal et al., 2012*; *Mehra et al., 2011*); (iv) CXR values higher than two on a scale of 0–4; (v) respiratory discomfort leading to vocalization; (vi) significant or complete loss of appetite; and (vii) detectable bacilli in BAL samples.

## Bacteria

Mtb H37Rv strain was grown in suspension in Middlebrook 7H9 medium (Difco) supplemented with 10% albumin-dextrose-catalase (ADC, Difco) and 0.05% Twen-80 (Sigma-Aldrich) (*Lastrucci et al., 2015*). For infection, growing Mtb was centrifuged (3000 rpm) at exponential phase stage and resuspended in PBS ($MgCl_2$, $CaCl_2$ free, Gibco). Twenty passages through a 26 G needle were done for dissociation of bacterial aggregates. Bacterial suspension concentration was then determined by measuring $OD_{600}$, and then resuspended in RPMI-1640 containing 10% FBS for infection.

## Viruses

Virus stocks were generated by transient transfection of 293 T cells with proviral plasmids coding for HIV-1 ADA and HIV-1 NLAD8-VSVG isolates, kindly provided by Serge Benichou (Institut Cochin, Paris, France), as previously described (*Vérollet et al., 2015*). Supernatant were harvested 2 days post-transfection and HIV-1 p24 antigen concentration was assessed by a homemade enzyme-linked immunosorbent assay (ELISA). HIV-1 infectious units were quantified, as reported (*Souriant et al., 2019*) using TZM-bl cells (NIH AIDS Reagent Program, Division of AIDS, NIAID, NIH from Dr. John C. Kappes, Dr. Xiaoyun Wu and Tranzyme Inc).

HIV-VLP stock (GFP VLP) was generated by transfecting the molecular clone pGag-eGFP obtained from the NIH AIDS Research and Reference Reagent Program. HEK-293 T cells were transfected with calcium phosphate (CalPhos, Clontech) in T75 flasks using 30 µg of plasmid DNA. Supernatants containing VLP were filtered (Millex HV, 0.45 µm; Millipore) and frozen at −80°C until use. The p24 Gag content of the VLP was determined by an ELISA (Perkin-Elmer).

## Preparation of human monocytes and monocyte-derived macrophages

Human monocytes were isolated from healthy subject (HS) buffy coat (from EFS) and differentiated towards macrophages as described (*Souriant et al., 2019*). Briefly, peripheral blood mononuclear cells (PBMC) were recovered by gradient centrifugation on Ficoll-Paque Plus (GE Healthcare). $CD14^+$ monocytes were then isolated by positive selection magnetic sorting, using human CD14 Microbeads and LS columns (Miltenyi Biotec). Cells were then plated at $1.6 \times 10^6$ cells per 6-well and allowed to differentiate for 5–7 days in RPMI-1640 medium (GIBCO), 10% Fetal Bovine Serum (FBS, Sigma-Aldrich) and human M-CSF (20 ng/mL) Peprotech) before infection with Mtb H37Rv for conditioned-media preparation. The cell medium was renewed every 3 or 4 days.

## Preparation of conditioned media

Conditioned-media from Mtb-infected macrophages (cmMTB) has been reported previously (*Lastrucci et al., 2015*; *Souriant et al., 2019*). Succinctly, hMDM were infected with Mtb H37Rv at a MOI of 3. After 18 hr of infection at 37°C, culture supernatants were collected, sterilized by double filtration (0.2 µm pores) and aliquots were stored at −80°C. We then tested the capacity of individual cmMTB to differentiate freshly isolated $CD14^+$ monocytes towards the M(IL-10) cell-surface marker phenotype, as assessed by FACS analyses. Those supernatants yielding a positive readout were then pooled together (5–10 donors) to minimize the inter-variability obtained between donors. Control media (cmCTR) was obtained from uninfected macrophage supernatant. When specified, IL-10 was eliminated from cmMTB by antibody depletion as described previously (*Lastrucci et al., 2015*;

*Souriant et al., 2019*). The depletion was verified by ELISA (BD-Bioscience), according to manufacturer's protocol.

## Conditioning of monocytes with the secretome of Mtb-infected macrophages or cytokines

Human CD14$^+$ sorted monocytes from HS buffy coat were allowed to adhere in the absence of serum (0.4 × 10$^6$ cells / 24-well in 500 μL) on glass coverslips, and then cultured for 3 days with 40% dilution (vol/vol) of cmCTR or cmMTB supplemented with 20% FBS and M-CSF (20 ng/mL, Peprotech). Blocking IFNAR receptor was performed by pre-incubation with mouse anti-IFNAR antibody (20 μg/mL, Thermo Fischer Scientific) in a 200 μL for 30 min prior to conditioning. After 3 days, cells were washed and collected for phenotyping.

When specified, monocytes were also conditioned in presence of 20 ng/mL M-CSF and 10 ng/mL recombinant human IL-10 (PeproTech) or 10 U/mL of IFNβ (Peprotech). Cell-surface expression of Siglec-1 was measured by flow cytometry using standard procedures detailed hereafter.

## RNA extraction and transcriptomic analysis

Cells conditioned with cmCTR and cmMTB supernatants (approximately 1.5 million cells) were treated with TRIzol Reagent (Invitrogen) and stored at −80℃. Total RNA was extracted from the TRIzol samples using the RNeasy mini kit (Qiagen). RNA amount and purity (absorbance at 260/280 nm) was measured with the Nanodrop ND-1000 apparatus (Thermo Scientific). According to the manufacturer's protocol, complementary DNA was then reverse transcribed from 1 μg total RNA with Moloney murine leukemia virus reverse transcriptase (Invitrogen), using random hexamer oligonucleotides for priming. The microarray analysis was performed using the Agilent Human GE 4 × 44 v2 (single color), as previously described (*Lugo-Villarino et al., 2018*). Briefly, we performed hybridization with 2 μg Cy3-cDNA and the hybridization kit (Roche NimbleGen). The samples were then incubated for 5 min at 65℃, and 5 min at 42℃ before loading for 17 hr at 42℃, according to manufacturer's protocol. After washing, the microarrays were scanned with MS200 microarray scanner (Roche NimbleGen), and using Feature Extraction software, the Agilent raw files were extracted and then processed through Bioconductor (version 3.1) in the R statistical environment (version 3.6.0). Using the limma package, raw expression values were background corrected in a 'normexp' fashion and then quantile normalized and log$_2$ transformed (*Ritchie et al., 2015*). Density plots, box-plots, principal component analyses, and hierarchical clustering assessed the quality of the hybridizations. Differentially expressed genes between macrophages exposed to cmCTR or cmMTB supernatants were extracted based on the p-value corrected using the Benjamini-Hochberg procedure. The log$_2$ normalized expression values were used to perform Gene Set Enrichment Analyses (GSEA). The GSEA method allows to statistically test whether a set of genes of interest (referred to as a geneset) is distributed randomly or not in the list of genes that were pre-ranked according to their differential expression ratio between macrophages exposed to cmCTR or cmMTB supernatants. The output of GSEA is a GeneSet enrichment plot. The vertical black lines represent the projection onto the ranked GeneList of the individual genes of the GeneSet. The top curve in green corresponds to the calculation of the enrichment score (ES). The more the ES curve is shifted to the upper left of the graph, the more the GeneSet is enriched in the red cell population. Conversely, the more the ES curve is shifted to the lower right of the graph, the more the GeneSet is enriched in the blue cell population.

## siRNA silencing

Targeted gene silencing in monocytes was performed using reverse transfection protocol as previously described (*Troegeler et al., 2014*). Shortly, human primary monocytes were transfected with 200 nM of ON-TARGETplus SMARTpool siRNA targeting Siglec-1 (Horizon Discovery) or non-targeting siRNA (control) using HiPerfect transfection system (Qiagen). After a four-hr post-transfection, cells were allowed to rest for 24 hr in RPMI-1640 medium, 10% FBS, 20 ng/mL of M-CSF, before addition of cmMTB media (40% vol/vol). After three additional days of conditioning, cells were infected with HIV-1 ADA or HIV-1- NLAD8-VSV-G strain, and kept in culture for 10 more days or 48 hrs, respectively. Validation of gene silencing was done after three days post-transfection, and this

protocol led to the efficient depletion of Siglec-1 ranging between 50 to95%, as measured by flow cytometry.

## HIV-1 infection

For HIV-1 infection, at day 3 of differentiation, $0.4 \times 10^6$ human monocytes-derived macrophages (hMDM) were infected with HIV-1 ADA strain (or as indicated otherwise) at MOI 0.1. HIV-1 infection, and replication were assessed at 10-day post-infection by measuring p24-positive cells by immunostaining and the level of p24 released in culture media by ELISA. For the infection and TNT quantification at day six post-infection, the same protocol was used. For HIV-1 transfer, higher MOI of HIV-1 VSVG pseudotyped NLAD8 virus was used, as described below (see section *HIV-1 and cell-to-cell transfer*) and in *Souriant et al. (2019)*.

## Uptake of Virus-Like particles

Uptake experiment were performed as previously described (*Izquierdo-Useros et al., 2012a*; *Izquierdo-Useros et al., 2014*; *Pino et al., 2015*) using p24$^{Gag}$ HIV-1$_{Gag-eGFP}$ VLP (GFP VLP). Briefly, monocytes transfected (or not) with control siRNA, or with siRNA directed against Siglec-1, and differentiated for 3 days in cmCTR or cmMTB, were washed once with PBS prior to addition of 2 ng/mL of GFP VLP. Binding was performed during 3.5 hr at 37°C in a 5% $CO_2$ incubator. Cells were then detached with cell dissociation buffer (Gibco) and prepared for flow cytometry analysis on a BD LSRFortessa (TRI-Genotoul platerform). Same experiment was also performed blocking monocyte-derived macrophages at RT for 15 min with 10 µg/mL of mAb α-Siglec-1 7–239 (Abcam), IgG1 isotype control (BD Biosciences) or leaving cells untreated before VLP addition.

## Flow cytometry and Siglec-1 quantitation

Staining of conditioned macrophages was performed as previously described (*Souriant et al., 2019*). Adherent cells were harvested after 5 min incubation in trypsin 0.05% EDTA (Gibco) and washes with PBS (Gibco). After 10 min centrifugation at 320 g, pellets were resuspended in cold staining buffer (PBS, 2 mM EDTA, 0.5% FBS) with fluorophore-conjugated antibodies (See Key ressources Table) and, in parallel, with the corresponding isotype control antibody using a general concentration of 1 µg/mL. After staining, cells were washed with cold staining buffer, centrifuged for 2 min at 320 g at 4°C, and analyzed by flow cytometry using BD LSRFortessa flow cytometer (BD Biosciences, TRI Genotoul plateform) and the associated BD FACSDiva software. Data were then analyzed using the FlowJo_V10 software (FlowJo, LLC). Gating on macrophage population was set according to its Forward Scatter (FSC) and Size Scatter (SSC) properties before doublet exclusion and analysis of the median fluorescence intensity (MFI) for each staining.

To determine Siglec-1 expression, we applied a quantitative FACS assay. Briefly, cmCTR- and cmMTB-treated macrophages were detached using Accutase solution (Gibco) for 10 min at 37°C, washed, blocked with 1 mg/mL human IgG (Privigen, Behring CSL), and stained with mAb 7–239 α-Siglec-1-PE or matched isotype-PE control (Biolegend) at 4°C for 30 min. The mean number of Siglec-1 mAb binding sites per cell was obtained with a Quantibrite kit (Becton Dickinson) as previously described (*Izquierdo-Useros et al., 2012b*). Samples were analyzed with FACSCalibur using CellQuest software to evaluate collected data.

## Immunofluorescence microscopy

Cells were fixed with PFA 3.7%, Sucrose 30 mM in PBS. After washing with PBS, cells were saturated with blocking buffer (PBS-BSA 1%). Depending on the experiments, cells were permeabilized as previously described (*Souriant et al., 2019*) with Triton X-100 0.3% for 10 min (or not), and then stained for 2 hrs with the primary antibody: anti-Siglec-1 (10 µg/mL, Novus Biologicals). Cells were then incubated with appropriate secondary antibodies for 1 hr: Alexa Fluor 488 or 555 or 647 Goat anti-Mouse IgG (2 µg/mL, Cell Signaling Technology). Cells were then permeabilized, washed in PBS before saturation with 0.6 µg/mL mouse IgG2 diluted in Dako Antibody Reducing Background buffer (Dako) for 30 min. Intracellular proteins were then stained with anti-Gag KC57 RD1 antibody (1/100, Beckman Coulter) and/or anti-α-tubulin (5 µg/mL, Abcam) for 2 hrs. Cells were washed and finally incubated with Alexa Fluor 488, 555 or 647 Goat anti-Mouse, or Goat anti-Rabbit IgG secondary antibodies (2 µg/mL, Cell Signaling Technology), Alexa Fluor 488 or 555 Phalloidin (33 mM, Thermo

Fisher Scientific), Wheat Germ Agglutinin (CF350 WGA, Thermofischer) and DAPI (500 ng/mL, Sigma Aldrich) in blocking buffer for 1 hr. Coverslips were mounted on a glass slide using Fluorescence Mounting Medium (Dako) and visualized with a spinning disk (Olympus), a Zeiss confocal LSM880 with Airyscan or a FV1000 confocal microscope (Olympus).

TNT were identified by WGA or phalloidin and tubulin staining, and counted on at least 1000 cells per condition and per donor.

As HIV-1 infection induces macrophages fusion into MGC (*Vérollet et al., 2010*), the number of infected cells largely underestimates the rate of infection. Thus, to better reflect the rate of infection, we quantified the percentage of MGC. Using semi-automatic quantification with homemade Image J macros, allowing the study of more than 5,000 cells per condition in at least five independent donors, we assessed these parameters.

### HIV-1 and cell-to-cell transfer

Freshly isolated CD14$^+$ monocytes from HS transfected with siRNA against Siglec-1 (or siRNA control) were allowed to adhere in the absence of serum ($2 \times 10^6$ cells/6-well in 1.5 mL). After 4 hr of culture, RPMI-1640 supplemented with 20 ng/mL M-CSF and 20% FBS were added to the cells (vol/vol). After 24 hr, cells were conditioned with cmMTB media. At day 4, 120 ng p24 of a HIV-1 NLAD8 strain pseudotyped with a VSVG envelope was used to infect half of the cells, kept in culture for two more days. At day 6, before co-culture, uninfected cells were stained with CellTracker Green CMFDA Dye (Thermo Fisher Scientific). For mitochondria transfer, half of the macrophages were pre-stained with Green CellTracker, and the other half, uninfected, was stained with mitoTracker Deep-Red prior to co-culture. Briefly, cells were washed with PBS Mg$^{2+}$/Ca$^{2+}$ and stained for 30 min with 500 ng/mL CellTracker or mitoTracker, before washing with RPMI-1640 10% FBS. HIV-1$^+$(or mitoTracker$^+$) and CellTracker$^+$ cells were then detached using accutase (Sigma) and co-cultured at a 1:1 ratio on glass coverslips in 24-well.

### Histological analyses

Paraffin embedded tissue samples were sectioned and stained with hematoxylin and eosin for histomorphological analysis. Different antigen unmasking methods where used on tissue slides for immunohistochemical staining, which was performed using anti-CD163 (Leica/Novocastra), anti-Siglec-1 (Novus Biologicals) and anti-pSTAT1 (Cell Signaling Technology). Sections were then incubated with biotin-conjugated polyclonal anti-mouse or anti-rabbit immunoglobulin antibodies, followed by the streptavidin-biotin-peroxidase complex (ABC) method (Vector Laboratories). Finally, sections were counter-stained with hematoxylin. Slides were scanned with the Panoramic 250 Flash II (3DHISTECH). Virtual slides were automatically quantified for macrophage distribution as previously described (*Souriant et al., 2019*). Immunofluorescence staining of the sections was performed as described above, and followed by anti-mouse IgG isotype specific or anti-rabbit IgG antibodies labelled with Alexa488 and Alexa555 (Molecular Probes). Samples were mounted with Prolong Antifade reagent (Molecular Probes) and examined using a 60x/1.40N.A. objective of an Olympus spinning disk microscope.

### Quantification and statistical analysis

Information on the statistical tests used and the exact values of n (donors) can be found in the Figure Legends. All statistical analyses were performed using GraphPad Prism 8.0.0 (GraphPad Software Inc). Two-tailed paired or unpaired t-test was applied on data sets with a normal distribution (determined using Kolmogorov-Smirnov test), whereas two-tailed Mann-Whitney (unpaired test) or Wilcoxon matched-paired signed rank tests were used otherwise. $p < 0.05$ was considered as the level of statistical significance (*$p \leq 0.05$; **$p \leq 0.005$; ***$p \leq 0.0005$; ****$p \leq 0.0001$).

## Acknowledgements

We greatly acknowledge F Capilla and T Al Saati, US006/CREFRE for histology analyses; P Constant, F Levillain, F Moreau and C Berrone, IPBS and Genotoul Anexplo-IPBS, for accessing the BSL3 facilities; E Näser, E Vega, A Peixoto, S Mazeres and the Genotoul TRI-IPBS facilities for imaging and flow cytometry. We thank F Quiroga and C del Carmen Melucci Ganzarain, Instituto de Investigaciones Biomédicas en Retrovirus y SIDA, INBIRS UBA - CONICET, Buenos Aires, Argentina, for the

technical help and advice provided. We greatly thank Y-M Boudehen for technical expertise provided in molecular biology, M Dalod and B Raynaud-Messina for fruitful discussions, and S Benichou for providing HIV-1 strains. We are grateful to D Hudrisier, C Gutierrez, CA Spinner, L Bernard-Raichon, and B Raymond for critical reading of the manuscript and helpful comments. This work was supported by the *Centre National de la Recherche Scientifique*, *Université Paul Sabatier*, the *Agence Nationale de la Recherche* (ANR16-CE13-0005-01, ANR-11-EQUIPEX-0003), the *Agence Nationale de Recherche sur le Sida et les Hépatites virales* ( ANRS2014-049, ANRS2018-01, ANRS2020-01), the ECOS-Sud program (A14S01), the *Fondation pour la Recherche Médicale* (DEQ2016 0334894; DEQ2016 0334902), INSERM Plan Cancer, Human Frontier Science Program (HFSP: RGP0035/2016), the Argentinean National Agency of Promotion of Science and Technology (PICT-2015–0055 and PICT-2017–1317). We also thank the AIDS Research and Reference Reagent Program, Division of AIDS, NIAID. The NHP study was supported by NIH award OD011104, AI111943, AI111914, AI097059 and AI110163 . The genetic analyses were realized within the framework of the Swiss HIV Cohort Study (SHCS Project number 717), which is supported by the Swiss National Science Foundation (Grant Number 148522) and by the SHCS research foundation. MD is supported by an ATP (*Axes Thématiques Prioritaires*) doctoral scholarship from *Université Paul Sabatier*, SS by a 4th-year doctoral scholarship from Sidaction, and S R by a scholarship from Toulouse University Hospital to perform a Master's degree. JM-P and NI-U are supported by the Spanish Secretariat of State of Research, Development and Innovation through grant SAF2016-80033-R, JM-P by the Spanish AIDS network *Red Temática Cooperativa de Investigación en SIDA*, and SB by the *Rio Hortega programme* funded by the Spanish Health Institute Carlos III (No. CM17/00242).

## Additional information

### Funding

| Funder | Grant reference number | Author |
| --- | --- | --- |
| Agence Nationale de Recherches sur le Sida et les Hépatites Virales | ANRS2014-049 | Isabelle Maridonneau-Parini<br>Olivier Neyrolles |
| Agence Nationale de Recherches sur le Sida et les Hépatites Virales | ANRS2020-01 | Christel Vérollet<br>Geanncarlo Lugo-Villarino |
| Agence Nationale de Recherches sur le Sida et les Hépatites Virales | ANRS2018-01 | Christel Vérollet |
| Agence Nationale de la Recherche | ANR-11-EQUIPEX-0003 | Olivier Neyrolles |
| Agence Nationale de la Recherche | ANR16-CE13-0005-01 | Christel Vérollet |
| Fondation pour la Recherche Médicale | DEQ20160334902 | Olivier Neyrolles |
| Fondation pour la Recherche Médicale | DEQ2016 0334894 | Isabelle Maridonneau-Parini |
| ECOS-Sud program | A14S01 | Maria del Carmen Sasiain<br>Olivier Neyrolles |
| Human Frontier Science Program | RGP0035/2016 | Isabelle Maridonneau-Parini |
| National Agency for Science and Technology, Argentina | PICT-2015–0055 | Maria del Carmen Sasiain |
| National Agency for Science and Technology, Argentina | PICT-2017–1317 | Luciana Balboa |
| National Institutes of Health | OD011104 | Deepak Kaushal<br>Marcelo J Kuroda |
| National Institutes of Health | AI111943 | Deepak Kaushal |

| | | |
|---|---|---|
| National Institutes of Health | AI097059 | Marcelo J Kuroda |
| National Institutes of Health | AI110163 | Marcelo J Kuroda |
| National Institutes of Health | AI111914 | Deepak Kaushal |
| Université Toulouse III - Paul Sabatier | Axes Thématiques Prioritaires doctoral scholarship | Maeva Dupont |
| Sidaction | 4th year doctoral scholarship | Shanti Souriant |
| Centre Hospitalier Universitaire de Toulouse | Master's scholarship | Stella Rousset |
| Ministerio de Economía y Competitividad | Secretariat of State of Research, Development and Innovation SAF2016-80033-R | Javier Martinez-Picado Nuria Izquierdo-Useros |
| Red Temática Cooperativa de Investigación en SIDA | | Javier Martinez-Picado |
| Institute of Health Carlos III | CM17/00242 | Susana Benet |

The funders had no role in study design, data collection and interpretation, or the decision to submit the work for publication.

## Author contributions

Maeva Dupont, Conceptualization, Data curation, Formal analysis, Investigation, Visualization, Methodology; Shanti Souriant, Formal analysis, Investigation, Visualization, Methodology; Luciana Balboa, Céline Cougoule, Susana Benet, Formal analysis, Investigation, Methodology; Thien-Phong Vu Manh, Resources, Data curation, Formal analysis, Investigation, Methodology; Karine Pingris, Renaud Poincloux, Investigation, Methodology; Stella Rousset, Data curation, Formal analysis, Investigation, Methodology; Yoann Rombouts, Conceptualization, Investigation, Methodology; Myriam Ben Neji, Data curation, Investigation, Methodology; Carolina Allers, Marcelo J Kuroda, Resources, Methodology; Deepak Kaushal, Resources, Investigation; Javier Martinez-Picado, Maria del Carmen Sasiain, Conceptualization, Resources, Project administration; Nuria Izquierdo-Useros, Resources, Formal analysis, Investigation, Methodology; Isabelle Maridonneau-Parini, Funding acquisition, Investigation; Olivier Neyrolles, Conceptualization, Funding acquisition, Investigation, Project administration; Christel Vérollet, Conceptualization, Formal analysis, Supervision, Funding acquisition, Validation, Investigation, Visualization, Methodology, Project administration; Geanncarlo Lugo-Villarino, Conceptualization, Formal analysis, Supervision, Funding acquisition, Validation, Investigation, Methodology, Project administration

## Author ORCIDs

Maeva Dupont (iD) https://orcid.org/0000-0002-0514-4447
Olivier Neyrolles (iD) https://orcid.org/0000-0003-0047-5885
Christel Vérollet (iD) https://orcid.org/0000-0002-1079-9085
Geanncarlo Lugo-Villarino (iD) https://orcid.org/0000-0003-4620-8491

## Ethics

Human subjects: Human Subjects Monocytes from healthy subjects were provided by Etablissement Français du Sang (EFS), Toulouse, France, under contract 21/PLER/TOU/IPBS01/20130042. According to articles L12434 and R124361 of the French Public Health Code, the contract was approved by the French Ministry of Science and Technology (agreement number AC 2009921). Written informed consents were obtained from the donors before sample collection.

Animal experimentation: Non-Human Primate (NHP) samples All animal procedures were approved by the Institutional Animal Care and Use Committee of Tulane University, New Orleans, LA and were performed at the Tulane TNPRC, and under approval from IACUC (project numbers P3733, P3794, P3373 and P3628). They were performed in strict accordance with NIH guidelines.

Decision letter and Author response
Decision letter https://doi.org/10.7554/eLife.52535.sa1
Author response https://doi.org/10.7554/eLife.52535.sa2

## Additional files

### Supplementary files
• Supplementary file 1. Clinical data of NHPs (Table S1) and histopathological scoring of lung lesions in NHPs (Table S2).

• Transparent reporting form

### Data availability
The raw data for the transcriptome analysis in this manuscript was made available through the public by a deposit to GEO under the accession code GSE139511.

The following dataset was generated:

| Author(s) | Year | Dataset title | Dataset URL | Database and Identifier |
|---|---|---|---|---|
| Mahn TPU, Geanncarlo L-V | 2020 | Tuberculosis-associated IFN-I induces Siglec-1 on microtubule-containing tunneling nanotubes and favors HIV-1 spread in macrophages | https://www.ncbi.nlm.nih.gov/geo/query/acc.cgi?acc=GSE139511 | NCBI Gene Expression Omnibus, GSE139511 |

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
