## [Decision Letter]

**Acceptance summary:**

Tuberculosis and HIV infection occur in lethal synergy, especially in regions such as Southern Africa and the collision between these two epidemics poses numerous public health and clinical management challenges. Despite the notable prevalence of such co-infections, the molecular basis of how the two disease causing agents combine to drive pathogenesis and further spread remains largely elusive. Your demonstration that the induction of Siglec-1, which occurs upon *Mycobacterium tuberculosis* infection, is associated with the transfer of HIV-1 to macrophages, through the use of tunnelling nanotubes (TNT), represents a notable advance for the field. This work creates new avenues for further delineating how these two pathogens modulate immunity. Further insight on this process will allow for the development of synergistic therapeutic options, that reduce mortality and paradoxic side-effects that current prevail in individuals who are co-infected with TB and HIV.

**Decision letter after peer review:**

Thank you for submitting your article "Tuberculosis-associated type I interferon induces Siglec-1 on tunneling nanotubes and favors HIV-1 spread in macrophages" for consideration by *eLife*. Your article has been reviewed by three peer reviewers, and the evaluation has been overseen by a Reviewing Editor and Päivi Ojala as the Senior Editor. The reviewers have opted to remain anonymous.

The reviewers have discussed the reviews with one another and the Reviewing Editor has drafted this decision to help you prepare a revised submission.

Summary:

The study by Dupont and colleagues describes a novel role for induction of Siglec-1, which occurs upon *Mycobacterium tuberculosis* infection, in the transfer of HIV-1 to macrophages through the use of tunnelling nanotubes (TNT). The authors have built upon an earlier article (Souriant et al., 2019) which demonstrated that the IL-10/STAT3 axis is involved in *M. tuberculosis*-mediated TNT formation and subsequent HIV-1 transfer in human macrophages.

Key findings:

1) Following identification of Siglec1 expression in transcriptomics experiments performed in macrophages treated with conditioned media, the authors demonstrated that this molecule is induced by type I IFNs and is associated with TNT formation as blocking via IFNAR2 abolishes the expression of Siglec-1.

2) They then find that Siglec-1 localizes on thick TNTs, associated with greater TNT length, HIV-1 and mitochondrial cargo. As a result, Siglec-1 appears to favour HIV-1 spread in macrophages.

3) The authors demonstrate that siRNA-mediated depletion of Siglec-1 significantly diminishes TNT transfer of HIV-1 from macrophages to recipient cells.

4) They then show that this has in vivo significance as the same phenomenon is detected in co-infected non-human primates.

Conclusion: Induction of Siglec-1 by tuberculosis infection is associated with the formation of TNTs that facilitate spread of HIV to macrophages, thus exacerbating disease.

Essential revisions:

1) Please confirm experimentally whether the effect seen with IFNAR inhibition is due to decreased levels of expression of Siglec-1 within macrophages or whether the percentage of Siglec-1+ cells is decreased.

2) Are Mtb-derived factors required for Siglec1-mediated HIV-1 transfer phenotype? An earlier study by some of the authors (Martinez-Picado et al., 2016) showed that lack of Siglec1 in HIV-1 infected patients does not have impact in HIV-1 acquisition or AIDS outcome. This original article has not been discussed in the present manuscript and could be important since the authors suggest Siglec1 as a new potential therapeutic target to limit viral dissemination. Please consider these findings and discuss them appropriately.

3) You demonstrated that Siglec-1 expression correlated positively with a greater length and high cargo of HIV-1 and mitochondria, arguing for a functional capacity of Siglec-1+ TNT to transfer material to recipient cells over long distances (subsection “Siglec-1 localization on thick TNT is associated with their length and HIV-1 cargo”). It is also stated that the "decreasing tendency for the capacity of Siglec-1-depleted cmMTB-treated macrophages to transfer mitochondria to recipient cells compared to controls (Figure 4E and Figure 4—figure supplement 1F), alludes to a possible defect in mechanisms involved in intercellular material transfer including through thick TNT". Given this function, one could envisage that Siglec-1 can alter cell-to-cell transfer of mitochondria by TNT, even irrespective of HIV-1 infection, altering the metabolism of the recipient cell. This point can be expanded in the Discussion.

4) Figure 2F and H; In panel H, it is clear that the percentages of Stat1+ cells in ATB-SIV are significantly higher than in either ATB or SIV macrophages. However, in panel F, it appears that the numbers of pSTAT1+ cells/mm2 is lower in ATB-SIV than in ATB. Please comment/address.

5) Figure 4—figure supplement 1. siRNA can have off target activity. If possible, controls to demonstrate specific siglec-1 depletion should be incorporated into the manuscript.

---

## [Author Response]

Essential revisions:1) Please confirm experimentally whether the effect seen with IFNAR inhibition is due to decreased levels of expression of Siglec-1 within macrophages or whether the percentage of Siglec-1+ cells is decreased.

We agree with the reviewers in pointing out the need to confirm this issue. First, it has been previously demonstrated that the IFNAR-2 blocking antibody reduces both the percentage of induction and level of expression of Siglec-1 triggered by the plasma of HIV-1-infected individuals in monocyte-derived dendritic cells (Pino et al., 2015). Second, as illustrated now in revised Figure 1—figure supplement 1D, the IFNAR-2 blocking antibody reduces the percentage of cmMTB-treated macrophages expressing Siglec-1, complementing the overall inhibition of the induction and level of expression of Siglec-1 in these cells (Figure 1C-D), as measured by flow cytometry analysis. Modifications to the revised manuscript have been annotated to incorporate Figure 1—figure supplement 1D. Last, we performed additional experiments to measure the intracellular levels of Siglec-1 expression in cmMTB-treated cells by an immunofluorescence approach. Results confirmed that, as opposed to cmCTR treatment, cmMTB increases the intracellular levels of Siglec-1 expression (revised Figure 1—figure supplement 1C). Upon blocking of the IFNAR-2, we observed a decreasing tendency in the percentage of cell expressing Siglec-1 intracellularly (Author response image 1). Collectively, these observations indicate that IFNAR inhibition leads to both decreased levels of expression (extra- and intracellularly) of Siglec-1 and a lower percentage of cells expressing this receptor as induced by cmMTB treatment.

**Author response image 1. respfig1:** Immunofluorescence analysis of cmMTB-treated cells positive for Siglec-1. For 3 days, monocytes were differentiated into macrophages either with cmCTR (white) or with cmMTB in the presence of an IFNAR-2 blocking (α-IFNAR, stripped) or control (α-IgG, black) antibodies. Immunofluorescence analyses were performed and the percentage of cells positive for intracellular Siglec-1 were quantified. n=4 donors.

2) Are Mtb-derived factors required for Siglec1-mediated HIV-1 transfer phenotype? An earlier study by some of the authors (Martinez-Picado et al., 2016) showed that lack of Siglec1 in HIV-1 infected patients does not have impact in HIV-1 acquisition or AIDS outcome. This original article has not been discussed in the present manuscript and could be important since the authors suggest Siglec1 as a new potential therapeutic target to limit viral dissemination. Please consider these findings and discuss them appropriately.

We thank the reviewers for raising this important issue. In the revised version of our manuscript, we now discuss the findings reported by Martinez-Picado et al.in 2016 in the context of our present study (Discussion, last paragraph). In fact, we previously discussed this issue in length, and we kindly invite the reviewers to read it (Martinez-Picado et al., 2017). Briefly, we previously described a loss-of-function variant in *SIGLEC-*1; a formal analysis of a large cohort of HIV-1-infected individuals identified homozygous and heterozygous subjects, whose cells were functionally null or partially defective for Siglec-1 activity in HIV-1 capture and transmission ex vivo. However, this analysis of the effect of Siglec-1 truncation on progression to AIDS was not conclusive based on the lack of statistical significance. Of note, this does not rule out a role of Siglec-1 in the progression of AIDS. As discussed in Martinez-Picado et al. 2017, there are multiple challenges that explain the lack of conclusive results in our approach to assess the impact of Siglec-1 genetic variants in AIDS progression in the human population. Briefly, the first challenge we encountered was the limited cohort size applied in our original study, which included 4’233 patients (Martinez-Picado et al., 2016). Indeed, power simulations determined that a proper analysis of a rare variant, like the Siglec-1 allele, would require more than 10,000 individuals to uncover a relative risk of 5 at P < 0.05 under a recessive model. Since the proposed effect demands long-term follow-up off therapy, it is very unlikely that a sufficient sample size could be obtained to determine the long-term consequences of the Siglec-1 stop variant on AIDS progression. Second, our original study lacked the seroconvertion date for the majority of the patients that were screened. This is an important issue because HIV-1 disease progression was only monitored from the diagnosis date, which may differ between individuals, especially if they are protected by the beneficial effect of the Siglec-1 stop variant. Third, there was a lack of complete clinical records from key individuals in our original study. For example, one of the homozygous individuals found for the loss-of-function variant in *SIGLEC-*1 had no clinical records for nine years; information for this individual only resumed after antiretroviral treatment initiation, when viral suppression could have masked any potential effect that the Siglec-1 variant might have had on disease progression. Last, but not least, the study of Siglec-1 genetic variants become more complex when considering potential infection by other pathogens that could abrogate any potential beneficial effects against AIDS progression. On this latter point, the reviewer is correct to infer direct influence from tuberculosis disease. For instance, one of the homozygous individuals for the rare Siglec-1 allele had a high CD4^+^ T cell count that dramatically dropped when tuberculosis was diagnosed, suggesting that the absence of a functional Siglec-1 receptor could have had a negative impact on the immune control of the mycobacterial infection. Indeed, in an independent study, we are currently addressing this issue where we suspect that Siglec-1 expression on myeloid APCs could compromise antigen capture *via* exosome or microvesicle transfer and affect the control of the *M. tuberculosis* infection (unpublished results). Therefore, these observations argue that the role of Siglec-1 in the progression of AIDS is still a pending issue, and that there is a strong need for future work targeting Siglec-1 to unveil the in vivo contribution of the mononuclear phagocyte system to HIV-1 pathogenesis to fully exploit the therapeutic potential of this receptor.

3) You demonstrated that Siglec-1 expression correlated positively with a greater length and high cargo of HIV-1 and mitochondria, arguing for a functional capacity of Siglec-1+ TNT to transfer material to recipient cells over long distances (subsection “Siglec-1 localization on thick TNT is associated with their length and HIV-1 cargo”). It is also stated that the "decreasing tendency for the capacity of Siglec-1-depleted cmMTB-treated macrophages to transfer mitochondria to recipient cells compared to controls (Figure 4E and Figure 4—figure supplement 1F), alludes to a possible defect in mechanisms involved in intercellular material transfer including through thick TNT". Given this function, one could envisage that Siglec-1 can alter cell-to-cell transfer of mitochondria by TNT, even irrespective of HIV-1 infection, altering the metabolism of the recipient cell. This point can be expanded in the Discussion.

We appreciate the reviewers’ suggestion. It is worth mentioning that decreasing tendency of mitochondrial transfer among cmMTB-treated cells depleted for Siglec-1 occurs in the absence of HIV-1 infection (Figure 4E). The fact that TNT-mediated mitochondria transfer is known to mediate metabolic changes in recipient cells, we intend to dissect whether Siglec-1 is responsible for modulating key energy-based metabolic pathways in recipient cells, such as aerobic glycolysis, pentose phosphate and lipid metabolism. This fascinating issue could have strong consequences to heath and disease beyond the infectious disease context. For this reason, as suggested by reviewers, we have now included a brief discussion in the revised text (Discussion, last paragraph).

4) Figure 2F and H; In panel H, it is clear that the percentages of Stat1+ cells in ATB-SIV are significantly higher than in either ATB or SIV macrophages. However, in panel F, it appears that the numbers of pSTAT1+ cells/mm2 is lower in ATB-SIV than in ATB. Please comment/address.

We agree with the reviewers in pointing out the need to confirm this issue. Originally, we decided to quantify specifically alveolar macrophages based on their location (within the alveolus), morphology (mononuclear cells with a large cytoplasm), and our previous characterization of these cells that are CD68^+^CD163^+^ (Lastrucci et al., 2015; Souriant et al., 2019). As the reviewers correctly point out, these quantifications were provided in the original Figure 1F (Siglec-1^+^ alveolar macrophages) and Figure 2H (pSTAT1^+^ alveolar macrophages) based on a single representative NHP. In addition, we also provided the quantification of both Siglec-1+ (original Figure 1G) and pSTAT-1 (original Figure 2F) found in the whole lung parenchyma tissue (per mm^2^) and based on a single representative NHP. On this latter point, although we previously determined most leukocytes are CD68^+^CD163^+^ (Lastrucci et al., 2015; Souriant et al., 2019), we did not expect all leukocytes to be only the macrophages in question, specifically for the pSTAT1 marker that is ubiquitously activated in most leukocytes during an infection context. Therefore, we were not surprised that pSTAT-1^+^ leukocytes (per mm^2^) were strongly present in the ATB group compared to ATB-SIV, as tuberculosis is well known to activate the IFN-I/STAT-1 signalling pathway (Moreira-Teixeira et al., 2018).

Nevertheless, we decided to now provide a more robust quantification analysis based on the cumulative number of alveolar macrophages that are positive for Siglec-1 (revised Figure 1F), pSTAT1 (revised Figure 2H, top), or both markers (revised Figure 2H, bottom), grouped from three independent animals for each NHP group. Moreover, we also provide the raw quantification for each image field containing the lung alveoli that was obtained from the three independent animals per NHP group to give the reviewer a general idea of the correlation found in alveolar macrophages that are positive for either Siglec-1 and pSTAT1 (Author response image 2). These analyses clearly confirmed that there is considerable increase of Siglec-1^+^pSTAT1^+^ alveolar macrophages in the co-infected NHP group compared to the others.

Likewise, we performed a similar quantification analysis of the cumulative number of lung parenchyma tissue cells that are positive for pSTAT1 (revised Figure 2F), and we still confirmed that lung parenchyma tissue Siglec-1^+^ leukocytes are highly abundant in the ATB group compared to the co-infected one. As the main message of our study is that a tuberculosis-associated environment leads to the upregulation of Siglec-1, and thus to susceptibility in the context of HIV-1 co-infection, we argue the results obtained in the lung parenchyma (in combination to those found in alveolar macrophages) are coherent and support our hypothesis.

**Author response image 2. respfig2:** Correlation of alveolar macrophages that are positive for either Siglec-1 or phospho-STAT1 in NHP (Related to Figure 2H, bottom). Each symbol represents an image field containing the lung alveoli that was obtained from the three independent animals per NHP group. Mean value is represented as a red line.

5) Figure 4—figure supplement 1. siRNA can have off target activity. If possible, controls to demonstrate specific siglec-1 depletion should be incorporated into the manuscript.

We appreciate the reviewers’ concern on this matter. First, we addressed this issue in our original research article where we optimized an siRNA-mediated gene silencing method in primary human macrophages, dendritic cells and monocytes (Troegeler et al., 2014). In this protocol, the used of ON-TARGETplus SMARTpool (according to Dharmarcon Inc instructions) for various genes were demonstrated to: i) specifically inactivate the desired gene of interest without affecting the expression of other related genes, ii) no effect on cell death, iii) no effect in the mononuclear phagocyte maturation and activation status, and iv) no effect on the mononuclear phagocyte capacity to phagocytose and migrate in different 3D environments. Second, the results obtained for Siglec-1 depletion by siRNA-mediated gene silencing were also validated using an alternate method based on blocking antibodies (see Figure 4—figure supplement 4D and E). Within this context, there is no cross-reaction of this monoclonal antibody with the closest molecules to Siglec-1, which are Siglec-5 and Siglec-7 (Perez-Zsolt et al., 2019, 10.1038/s41564-019-0453-2). Finally, we decided to measure other markers that are important to characterized the M(IL-10) phenotype induced by the cmMTB-treated in cells depleted for Siglec-1 by siRNA-mediated gene silencing. As illustrated in Author response image 3, while Siglec-1 expression is reduced cmMTB-treated cells from three independent donors, the expression of the CD16 and CD163 markers remained unchanged. Altogether, these results indicate that our siRNA-mediated gene silencing method applied to reduce the expression of Siglec-1 is unlikely to have off-target activity.

**Author response image 3. respfig3:** Inactivation of Siglec-1 by siRNA-mediated gene silencing does not affect the expression of M(IL-10) markers in cmMTB-treated cells (Related to Figure 4—figure supplement 1A). Monocytes from healthy subjects were transfected with siRNA targeting of Siglec-1 (siSiglec-1, black) or not (siCtrl, white). A day after, monocytes were differentiated into macrophages with cmMTB for 3 days. Vertical scatter plots showing the median fluorescent intensity (MFI, upper panels) of Siglec-1, CD16 and CD163, and the percentage of cells expressing Siglec-1, CD16 and CD163 (lower panels). Each circle represents a single donor and histograms median values. n=3 donors.